# Implicit Regularization of AdaDelta

**Matthias Englert**[*]                                   *m.englert@warwick.ac.uk*
*University of Warwick*

**Ranko Lazić**[*]                                        *r.s.lazic@warwick.ac.uk*
*University of Warwick*

**Avi Semler**[*]                                         *avi.semler@warwick.ac.uk*
*University of Warwick*

**Reviewed on OpenReview:** *https: // openreview. net/ forum? id= nm4OlbbwoR*

## Abstract

We consider the AdaDelta adaptive optimization algorithm on locally Lipschitz, positively homogeneous, and o-minimally definable non-smooth neural networks, with either the exponential or the logistic loss. We prove that, after achieving perfect training accuracy, the resulting adaptive gradient flows converge in direction to a Karush-Kuhn-Tucker point of the margin maximization problem, i.e. perform the same implicit regularization as the plain gradient flows. We also prove that the loss decreases to zero and the Euclidean norm of the parameters increases to infinity at the same rates as for the plain gradient flows. Moreover, we consider generalizations of AdaDelta where the exponential decay coefficients may vary with time and the numerical stability terms may be different across the parameters, and we obtain the same results provided the former do not approach 1 too quickly and the latter have isotropic quotients. Finally, we corroborate our theoretical results by numerical experiments on convolutional networks with MNIST and CIFAR-10 datasets.

## 1   Introduction

Understanding when, why, and how training overparameterized neural networks by gradient-based algorithms achieves good generalization (Zhang, Bengio, Hardt, Recht, and Vinyals, 2021; Belkin, Hsu, Ma, and Mandal, 2019) remains one of the central questions in machine learning, despite several years of vibrant research. Much progress on the question has been made by investigating *implicit regularization* (or implicit bias) (Neyshabur, Bhojanapalli, McAllester, and Srebro, 2017): the mysterious preference of the training algorithms for interpolators that perform well at test time.

Buiding on the seminal work of Soudry, Hoffer, Nacson, Gunasekar, and Srebro (2018), one of the most celebrated results in the field was obtained by Lyu and Li (2020); Ji and Telgarsky (2020): that after achieving perfect training accuracy, gradient flow implicitly regularizes locally Lipschitz, positively homogenous, and o-minimally definable non-smooth networks so that their parameters converge in direction to a margin maximization KKT point. This precise bias towards margin maximization for this wide class of networks has been the basis of numerous theoretical works (cf. Vardi (2023)), as well as remarkable practical methods such as the reconstruction of training data by Haim, Vardi, Yehudai, Shamir, and Irani (2022); Buzaglo, Haim, Yehudai, Vardi, Oz, Nikankin, and Irani (2023).

Nevertheless, already Soudry et al. (2018) observed that algorithms such as Adam (Kingma and Ba, 2015) may not perform the same implicit regularization, and posed the research question:

> *Can we characterize the bias of adaptive optimization algorithms for classification problems?*

---

[*]Equal contribution.

The pertinence of this question was further attested by Wilson, Roelofs, Stern, Srebro, and Recht (2017), who demonstrated that, for several realistic deep learning models, the solutions found by adaptive methods often generalize significantly worse than those found by stochastic gradient descent, even when the former solutions have better training performance.

Extending the pioneering work of Gunasekar, Lee, Soudry, and Srebro (2018); Qian and Qian (2019); Zhou, Feng, Ma, Xiong, Hoi, and E (2020), a major advance was made by Wang, Meng, Chen, and Liu (2021) who proved that, for the same wide class of networks as admitted by Lyu & Li (2020); Ji & Telgarsky (2020) and in the continuous setting corresponding to an infinitesimal learning rate, if the adapter[1] of an algorithm without momentum can be shown to converge to an isotropic vector without large fluctuations, then the implicit regularization is the same as for plain gradient flow. They also established that this holds for RMSProp (Hinton, Srivastava, and Swersky, 2012) and Adam without momentum,[2] but fails for AdaGrad (Duchi, Hazan, and Singer, 2011).

**Our contributions.** In this work, we tackle the posed research question for AdaDelta (Zeiler, 2012), which has remained open. AdaDelta is one of the main adaptive optimization algorithms, implemented in PyTorch[3], and known to perform well in many circumstances compared to other algorithms including RMSProp and Adam (cf. e.g. Ruder (2016)). Specifically:

- we overcome the technical challenge of AdaDelta having exponentially decaying averages in both the numerator and the denominator of the adapter (which makes it not readily amenable to the techniques of Wang et al. (2021)), and prove that the adapter has the required convergence properties, which enables us to conclude the same implicit regularization as for plain gradient flow;

- we show that the implicit regularization critically depends on the numerical stability terms in the numerator and the denominator of the adapter, and that it changes if they are permitted to have different components so that their quotient is not isotropic;

- we also investigate permitting the exponential decay coefficients to vary with time, and show that under a mild assumption on their integrals, the implicit regularization is not affected;

- we corroborate these theoretical results in three empirical settings, ranging from a simple visualization to a 14-layer convolutional network on CIFAR-10.

**Further related work.** The implicit regularization of adaptive optimization algorithms was the focus of several other recent works. Wang, Meng, Zhang, Sun, Chen, Ma, and Liu (2022) proved that momentum does not affect the implicit bias towards margin maximization for linear classification. Tarzanagh, Li, Zhang, and Oymak (2023) showed margin maximization when optimizing attention by gradient descent. Cattaneo, Klusowski, and Shigida (2024) studied the implicit bias of Adam and RMSProp by backward error analysis. Zhang, Zou, and Cao (2024) established that Adam without a numerical stability term maximizes the $\infty$-norm margin for linear logistic regression. Some of the prior empirical works on generalization of adaptive methods are Keskar and Socher (2017); Zaheer, Reddi, Sachan, Kale, and Kumar (2018); Luo, Xiong, Liu, and Sun (2019); Chen, Zhou, Tang, Yang, Cao, and Gu (2020).

Convergence properties of the popular Adam algorithm have been considered further in works that include the following. Wang, Fu, Zhang, Zheng, and Chen (2023) proved a tight upper bound on Adam's iteration complexity. Wang, Zhang, Zhang, Meng, Sun, Ma, Liu, Luo, and Chen (2024) studied convergence of randomly reshuffled Adam with diminishing learning rate and without assuming bounded smoothness, and showed that it can be faster than stochastic gradient descent with diminishing learning rate. Thilak, Littwin, Zhai, Saremi, Paiss, and Susskind (2024) identified and investigated a slingshot phenomenon in late-stage Adam and related it to grokking (Power, Burda, Edwards, Babuschkin, and Misra, 2022). Xie and Li (2024) showed that Adam with decoupled weight decay implicitly performs constrained optimization.

---

[1]In an adaptive algorithm, for each model parameter, the learning rate is adapted by a separate coefficient. The "adapter" consists of those coefficients, and is updated at every step of the algorithm.

[2]Here the difference between RMSProp and Adam without momentum is that the latter involves a bias-correction coefficient in the computation of the exponentially decaying average.

[3]https://pytorch.org/docs/stable/generated/torch.optim.Adadelta.html

A number of works have investigated convergence of classes of adaptive optimization algorithms with momentum, often by means of continuous-time limits. Da Silva and Gazeau (2020) derived a system of non-autonomous differential equations that is the limit for a class of algorithms that includes Adam, AdaFom, Heavy Ball, and Nesterov's Accelerated Gradient (NAG); and they studied convergence of its trajectories. In a concurrent work, Barakat and Bianchi (2021) focused on a similar system of equations, handling an irregular vector field that arises from the original version of Adam, and establishing further connections with the discrete-time iterates. The latter were extended by Barakat, Bianchi, Hachem, and Schechtman (2021) to allow noise in the Euler discretization of the non-autonomous differential equation system, thus being able to express algorithms such as stochastic Heavy Ball and stochastic NAG, and in particular to prove convergence of stochastic NAG in a non-convex setting. In a concurrent work, Gadat and Gavra (2022) were also able to prove almost sure escape of local maxima with low mini-batch noise level. More recently, Xiao, Hu, Liu, and Toh (2024) provided a framework for establishing convergence properties of Adam-family algorithms without assuming smoothness, which is thus applicable to neural networks with non-smooth activations.

Understanding generalization in deep learning by means of obtaining bounds in terms of the normalized margin and related quantities has attracted extensive attention, cf. e.g. Bartlett, Foster, and Telgarsky (2017); Neyshabur, Bhojanapalli, and Srebro (2018); Golowich, Rakhlin, and Shamir (2018); Wei, Lee, Liu, and Ma (2019); Wei and Ma (2020); Glasgow, Wei, Wootters, and Ma (2023). Although a variety of questions remain open, there is significant evidence behind the general understanding that larger normalized margins lead to better generalization, cf. e.g. (Jiang, Krishnan, Mobahi, and Bengio, 2019; Jiang, Neyshabur, Mobahi, Krishnan, and Bengio, 2020).

## 2 Preliminaries

**Basic notation.** We write: $[n]$ for the set $\{1, \ldots, n\}$; $\langle \boldsymbol{u}, \boldsymbol{v} \rangle$ for the inner product of vectors $\boldsymbol{u}$ and $\boldsymbol{v}$; $\|\boldsymbol{v}\| = \sqrt{\langle \boldsymbol{v}, \boldsymbol{v} \rangle}$ for the Euclidean length of a vector $\boldsymbol{v}$; $\boldsymbol{v}_i$ for the $i$th component of a vector $\boldsymbol{v}$; $\boldsymbol{0}, \boldsymbol{1}, \boldsymbol{\infty}$, etc. for the vectors whose dimension is inferred from the context and whose all components are equal to the specified value, so that for all $i$ we have $\boldsymbol{0}_i = 0$, $\boldsymbol{1}_i = 1$, $\boldsymbol{\infty}_i = \infty$, etc.

Provided $\boldsymbol{u}$ and $\boldsymbol{v}$ are vectors of equal dimensions, we write $\boldsymbol{uv}$, $\boldsymbol{u}/\boldsymbol{v}$, $\boldsymbol{v}^2$, $\sqrt{\boldsymbol{v}}$, etc. for the component-wise product, quotient, square, square root, etc. operations, so that for all $i$ we have $(\boldsymbol{uv})_i = \boldsymbol{u}_i \boldsymbol{v}_i$, $(\boldsymbol{u}/\boldsymbol{v})_i = \boldsymbol{u}_i/\boldsymbol{v}_i$, $(\boldsymbol{v}^2)_i = (\boldsymbol{v}_i)^2$, $(\sqrt{\boldsymbol{v}})_i = \sqrt{\boldsymbol{v}_i}$, etc. Similarly, we write $\boldsymbol{u} < \boldsymbol{v}$, $\boldsymbol{u} \leq \boldsymbol{v}$, etc. for the component-wise less, less than or equal, etc. relations, so that we have $\boldsymbol{u} < \boldsymbol{v} \Leftrightarrow \forall i\colon \boldsymbol{u}_i < \boldsymbol{v}_i$, $\boldsymbol{u} \leq \boldsymbol{v} \Leftrightarrow \forall i\colon \boldsymbol{u}_i \leq \boldsymbol{v}_i$, etc.

**Local Lipschitz continuity and the Clarke subdifferential.** Suppose a function $f\colon \mathbb{R}^k \to \mathbb{R}$ is *locally Lipschitz*, i.e. every point $\boldsymbol{v} \in \mathbb{R}^k$ has a neighborhood $U$ such that $f$ is Lipschitz continuous on $U$. By Rademacher's theorem (cf. e.g. Borwein and Lewis (2010, Theorem 9.1.2)), then $f$ is differentiable almost everywhere. The *Clarke subdifferential* of $f$ at a point $\boldsymbol{v}$ is the convex hull

$$\partial f(\boldsymbol{v}) := \operatorname{conv} \left\{ \lim_{i \to \infty} \nabla f(\boldsymbol{v}^{(i)}) \;\middle|\; \lim_{i \to \infty} \boldsymbol{v}^{(i)} = \boldsymbol{v} \text{ and } \nabla f(\boldsymbol{v}^{(i)}) \text{ exists for all } i \right\}.$$

It is nonempty and compact for all $\boldsymbol{v}$, and equals the singleton $\{\nabla f(\boldsymbol{v})\}$ if $f$ is continuously differentiable at $\boldsymbol{v}$ (Clarke, 1975). It consists of *subgradients*, which we may refer to simply as gradients.

**O-minimal structures and definable functions.** An *o-minimal structure* $\mathcal{S}$ is a family $\{\mathcal{S}_k\}_{k=1}^{\infty}$ such that: each $\mathcal{S}_k$ is a set of subsets of $\mathbb{R}^k$; $\mathcal{S}_1$ is the set of all finite unions of open intervals and points; each $\mathcal{S}_k$ contains the zero sets of all polynomials on $\mathbb{R}^k$; each $\mathcal{S}_k$ is closed under finite union, finite intersection, and complement; each $\mathcal{S}_{k+k'}$ contains the Cartesian products of all sets in $\mathcal{S}_k$ and $\mathcal{S}_{k'}$; each $\mathcal{S}_k$ contains the projections of all sets in $\mathcal{S}_{k+1}$ onto the first $k$ components. A function $f\colon \mathbb{R}^k \to \mathbb{R}^{k'}$ is *definable* in $\mathcal{S}$ if and only if its graph is a set in $\mathcal{S}_{k+k'}$.

For every o-minimal structure, the collection of all definable functions is closed under algebraic operations, composition, inverse, maximum, minimum, etc. (cf. e.g. Ji & Telgarsky (2020, Appendix B)). Moreover, by Wilkie's theorem (Wilkie, 1996), there exists an o-minimal structure in which the exponential function is definable.

---

**Procedure 1** *Discrete generalized AdaDelta.* In step $k+1$, using a current gradient $\widetilde{\partial}\mathcal{L}(\boldsymbol{w}_k) \in \partial\mathcal{L}(\boldsymbol{w}(t))$, it computes the next exponentially decaying average $\boldsymbol{g}_{k+1}$ of squared gradients, and computes the next adapted gradient $\boldsymbol{\Delta}_{k+1}$. It then computes the next exponentially decaying average $\boldsymbol{h}_{k+1}$ of squared adapted gradients, and computes the next parameter vector $\boldsymbol{w}_{k+1}$ by subtracting $\boldsymbol{\Delta}_{k+1}$ scaled by the learning rate. The hyperparameters are: learning rate $\eta > 0$, exponential decay coefficients $\boldsymbol{\rho}_k \in [0,1]^p$, and numerical stability terms $\boldsymbol{\delta} > \boldsymbol{0}$ and $\boldsymbol{\varepsilon} > \boldsymbol{0}$. The implementation of AdaDelta in PyTorch[3] corresponds to the special case where $\boldsymbol{\rho}_k = \varrho\,\boldsymbol{1}$ for all $k$, and $\boldsymbol{\delta} = \boldsymbol{\varepsilon} = \epsilon\,\boldsymbol{1}$. By specializing further to $\eta = 1$, we obtain the original AdaDelta (Zeiler, 2012).

$$\boldsymbol{g}_{k+1} = \boldsymbol{\rho}_k \boldsymbol{g}_k + (1 - \boldsymbol{\rho}_k)\widetilde{\partial}\mathcal{L}(\boldsymbol{w}_k)^2$$
$$\boldsymbol{h}_{k+1} = \boldsymbol{\rho}_k \boldsymbol{h}_k + (1 - \boldsymbol{\rho}_k)\boldsymbol{\Delta}_{k+1}^2 \qquad \boldsymbol{\Delta}_{k+1} = \sqrt{\frac{\boldsymbol{\varepsilon} + \boldsymbol{h}_k}{\boldsymbol{\delta} + \boldsymbol{g}_{k+1}}}\,\widetilde{\partial}\mathcal{L}(\boldsymbol{w}_k)$$
$$\boldsymbol{w}_{k+1} = \boldsymbol{w}_k - \eta\boldsymbol{\Delta}_{k+1}$$

---

**Predictor and loss functions.** We assume the following properties of the predictor function $\Phi(\boldsymbol{w}, \boldsymbol{x})$ with parameters $\boldsymbol{w} \in \mathbb{R}^p$, inputs $\boldsymbol{x} \in \mathbb{R}^d$, and scalar outputs. The $L$-positive homogeneity means that scaling the parameters $\boldsymbol{w}$ by any $\alpha > 0$ scales the output by $\alpha^L$, i.e. $\Phi(\alpha\boldsymbol{w}, \boldsymbol{x}) = \alpha^L \Phi(\boldsymbol{w}, \boldsymbol{x})$.

**Assumption 1.** *For some $L > 0$ and some o-minimal structure $\mathcal{S}$ in which the exponential function is definable, for each $\boldsymbol{x}$ we have that $\Phi(\boldsymbol{w}, \boldsymbol{x})$ as a function of $\boldsymbol{w}$ is: (i) locally Lipschitz; (ii) $L$-positively homogeneous; (iii) definable in $\mathcal{S}$.*

This assumption admits neural networks that are constructed from a wide variety of layer types, including fully connected, convolutional, ReLU, Leaky ReLU, Square ReLU, cube activation, and max-pooling, which may be composed arbitrarily. Points of nondifferentiability, such as at 0 in the case of the ReLU nonlinearity, are permitted because we assume only local Lipschitzness instead of continuous differentiability and we work with the Clarke subdifferential instead of the gradient. However, due to the $L$-positive homogeneity (ii), skip connections are excluded, biases are excluded except at the first layer, and activations based on the exponential function (e.g. sigmoid activation) are excluded. The homogeneity exponent $L$ is the degree to which the parameters are multiplied when computing the network output — this may be determined by starting with exponent 0, then going forwards through the layers and e.g.: adding 1 to the exponent if the layer is fully connected, not changing the exponent if the layer is ReLU, doubling the exponent if the layer is Square ReLU, etc.

We consider minimizing the total loss $\mathcal{L}(\boldsymbol{w}) := \sum_{i=1}^n \ell\left(y^{(i)}\Phi(\boldsymbol{w}, \boldsymbol{x}^{(i)})\right)$, which is with respect to a finite dataset $\left\{(\boldsymbol{x}^{(i)} \in \mathbb{R}^d, y^{(i)} \in \{\pm 1\})\right\}_{i=1}^n$ and where the individual loss function is one of:

$$\ell(z) := \begin{cases} e^{-z} & \text{for the exponential loss;} \\ \log(1 + e^{-z}) & \text{for the logistic loss.} \end{cases}$$

Now we make use of the definability of the exponential function in the o-minimal structure $\mathcal{S}$ from Assumption 1. (The exponential function was effectively excluded by the homogeneity clause (ii) from the building of the predictor function $\Phi(\boldsymbol{w}, \boldsymbol{x})$.) Since the individual loss function $\ell$ (in both cases, exponential and logistic) is locally Lipschitz and definable in $\mathcal{S}$, the same is true of the total loss function $\mathcal{L}$. Note however that the loss functions we consider are not homogeneous.

**Generalized AdaDelta flow trajectories.** The starting point of our analysis is the adaptive gradient descent of Procedure 1, which is a generalization of AdaDelta (Zeiler, 2012) by allowing: the learning rate to be specified (this is already the case in the PyTorch implementation[3]), the exponential decay coefficients to vary with time (e.g. by following a specified schedule), and both those coefficients and the numerical stability terms to be specified differently across the vector components.

The focus of our theoretical study is the adaptive gradient flow that corresponds to the adaptive gradient descent with an infinitesimal learning rate. To arrive at its definition, we first restate the equations of Procedure 1 as follows. We eliminate the auxiliary variable $\boldsymbol{\Delta}_{k+1}$, we suppose the hyperparameters $\boldsymbol{\rho}_k$ obey

---

**Process 2** *Continuous generalized AdaDelta.* This adaptive gradient flow, where $\widetilde{\partial}\mathcal{L}(\boldsymbol{w}(t)) \in \partial\mathcal{L}(\boldsymbol{w}(t))$ is a gradient of the loss at the current parameters $\boldsymbol{w}(t)$, corresponds to the adaptive gradient descent of Procedure 1 with an infinitesimal learning rate. The hyperparameters are: exponential decay coefficients schedule $\boldsymbol{\rho}\colon [0,\infty) \to [0,1]^p$, and numerical stability terms $\boldsymbol{\delta} > \boldsymbol{0}$ and $\boldsymbol{\varepsilon} > \boldsymbol{0}$.

$$\boldsymbol{g}'(t) = (\boldsymbol{1} - \boldsymbol{\rho}(t))(\widetilde{\partial}\mathcal{L}(\boldsymbol{w}(t))^2 - \boldsymbol{g}(t)) \tag{1}$$

$$\boldsymbol{h}'(t) = (\boldsymbol{1} - \boldsymbol{\rho}(t))(\boldsymbol{w}'(t)^2 - \boldsymbol{h}(t)) \tag{2}$$

$$\boldsymbol{w}'(t) = -\sqrt{\frac{\boldsymbol{\varepsilon} + \boldsymbol{h}(t)}{\boldsymbol{\delta} + \boldsymbol{g}(t)}}\, \widetilde{\partial}\mathcal{L}(\boldsymbol{w}(t)) \tag{3}$$

---

that $(\boldsymbol{1} - \boldsymbol{\rho}_k)/\eta$ is constant with respect to the hyperparameter $\eta$, and we replace steps $k$ by times $t = k\eta$:

$$\frac{\boldsymbol{g}(t+\eta) - \boldsymbol{g}(t)}{\eta} = (\boldsymbol{1} - \boldsymbol{\rho}(t))\left(\widetilde{\partial}\mathcal{L}(\boldsymbol{w}(t))^2 - \boldsymbol{g}(t)\right)$$

$$\frac{\boldsymbol{h}(t+\eta) - \boldsymbol{h}(t)}{\eta} = (\boldsymbol{1} - \boldsymbol{\rho}(t))\left(\left(\frac{\boldsymbol{w}(t+\eta) - \boldsymbol{w}(t)}{\eta}\right)^2 - \boldsymbol{h}(t)\right)$$

$$\frac{\boldsymbol{w}(t+\eta) - \boldsymbol{w}(t)}{\eta} = -\sqrt{\frac{\boldsymbol{\varepsilon} + \boldsymbol{h}(t)}{\boldsymbol{\delta} + \boldsymbol{g}(t+\eta)}}\, \widetilde{\partial}\mathcal{L}(\boldsymbol{w}(t))\,.$$

Now, regarding these equations as determining the endpoints $\boldsymbol{g}(t+\eta), \boldsymbol{h}(t+\eta), \boldsymbol{w}(t+\eta)$ of the next line segments in some continuous polygonal curves $\boldsymbol{g}, \boldsymbol{h}, \boldsymbol{w}\colon [0,\infty) \to \mathbb{R}^p$ born from the adaptive gradient descent of Procedure 1 with learning rate $\eta$ (cf. e.g. Elkabetz and Cohen (2021)), letting $\eta$ tend to 0 we obtain that the limits of their directions are given by the right-hand sides of eqs. (1) to (3), which we take to be the derivatives that determine the adaptive gradient flow we are seeking.

We therefore analyze *trajectories* $\boldsymbol{g}, \boldsymbol{h}, \boldsymbol{w}\colon [0,\infty) \to \mathbb{R}^p$ of the two exponentially decaying averages and of the parameters, which are arcs (i.e. absolutely continuous on every compact subinterval) and which obey the adaptive gradient flow of Process 2 for almost all $t \geq 0$.

**Assumption 2.** *(i)* $\int_0^\infty (\boldsymbol{1} - \boldsymbol{\rho}(t))\mathrm{d}t = \infty$. *(ii)* $\boldsymbol{g}(0) \geq \boldsymbol{0}$ *and* $\boldsymbol{h}(0) \geq \boldsymbol{0}$. *(iii) There exists a time $t_0$ such that* $\mathcal{L}(\boldsymbol{w}(t_0)) < \ell(0)$.

This is a mild regularity assumption. Part (i) ensures that the exponential decay coefficients are not scheduled to approach 1 so fast that they stop the learning, and for example it is satisfied by any constant coefficients smaller than 1. Part (ii) just requires that the exponentially decaying averages of squared gradients and squared adapted gradients are initialized as nonnegative (in original AdaDelta they are initialized to zero). Part (iii) assumes that the dataset is separable by the network, moreover that for every parameters trajectory $\boldsymbol{w}$ that we consider arising from the generalized AdaDelta flow, there exists a time $t_0$ at which a separation (i.e. the perfect training accuracy) is achieved; this commonly occurs when training overparameterized neural networks by gradient based algorithms (cf. e.g. Zhang et al. (2021)). We also remark that, for both the exponential and the logistic individual loss function $\ell(z)$, if $f(z)$ is defined by $\ell(z) = \mathrm{e}^{-f(z)}$, then $f(z)$ is monotonically increasing, and so the condition $\mathcal{L}(\boldsymbol{w}(t_0)) < \ell(0)$ is equivalent to $f^{-1}(\log(1/\mathcal{L}(\boldsymbol{w}(t_0)))) > 0$.

**Admittance of a chain rule.** Since the total loss function $\mathcal{L}$ is locally Lipschitz and definable in $\mathcal{S}$ (for both the exponential and the logistic individual loss functions), and the trajectory $\boldsymbol{w}$ of the parameters is an arc, we are able to use the following fact applied to them.

**Proposition 1** (Davis, Drusvyatskiy, Kakade, and Lee (2020, Theorem 5.8)). *If $f\colon \mathbb{R}^k \to \mathbb{R}$ is locally Lipschitz and definable in an o-minimal structure, then it admits a chain rule: for all arcs $\boldsymbol{v}\colon [0,\infty) \to \mathbb{R}^k$, almost all $t \geq 0$, and all $\boldsymbol{u} \in \partial f(\boldsymbol{v}(t))$, we have $\mathrm{d}f(\boldsymbol{v}(t))/\mathrm{d}t = \langle \boldsymbol{u}, \mathrm{d}\boldsymbol{v}(t)/\mathrm{d}t \rangle$.*

**Directional convergence, KKT conditions, and margin maximization.** That a trajectory $\boldsymbol{v}$ *converges in direction* to a vector $\boldsymbol{u}$ means $\lim_{t\to\infty} \boldsymbol{v}(t)/\|\boldsymbol{v}(t)\| = \boldsymbol{u}/\|\boldsymbol{u}\|$.

Following Dutta, Deb, Tulshyan, and Arora (2013, Section 2.2), supposing $f, g_1, \ldots, g_n : \mathbb{R}^k \to \mathbb{R}$ are locally Lipschitz, we have that $\boldsymbol{v} \in \mathbb{R}^k$ is a *Karush-Kuhn-Tucker point* of the problem

$$\text{minimize: } f(\boldsymbol{v}) \qquad \text{subject to: } g_i(\boldsymbol{v}) \leq 0 \text{ for all } i \in [n]$$

if and only if there exist Lagrange multipliers $\lambda_1, \ldots, \lambda_n \geq 0$ such that:

**(feasibility)** $g_i(\boldsymbol{v}) \leq 0$ for all $i \in [n]$;

**(equilibrium inclusion)** $\boldsymbol{0} \in \partial f(\boldsymbol{v}) + \sum_{i=1}^n \lambda_i \partial g_i(\boldsymbol{v})$;

**(complementary slackness)** $\lambda_i g_i(\boldsymbol{v}) = 0$ for all $i \in [n]$.

For local optimality, these first-order conditions are necessary, however in general not sufficient.

The *margin* of a parameters vector $\boldsymbol{w}$ is the smallest label-adjusted prediction for an input, i.e. formally $\min_{i \in [n]} y^{(i)} \Phi(\boldsymbol{w}, \boldsymbol{x}^{(i)})$. By the $L$-positive homogeneity of the predictor function (cf. Assumption 1.(ii)), the *normalized margin*

$$\min_{i \in [n]} y^{(i)} \Phi(\boldsymbol{w}, \boldsymbol{x}^{(i)}) / \|\boldsymbol{w}\|^L = \min_{i \in [n]} y^{(i)} \Phi(\boldsymbol{w}/\|\boldsymbol{w}\|, \boldsymbol{x}^{(i)})$$

depends only on the direction of $\boldsymbol{w}$, and it is straightforward to show that the directions that maximize it are also the optimal directions of the problem

$$\text{minimize: } \tfrac{1}{2}\|\boldsymbol{w}\|^2 \qquad \text{subject to: } y^{(i)} \Phi(\boldsymbol{w}, \boldsymbol{x}^{(i)}) \geq 1 \text{ for all } i \in [n] \,. \tag{4}$$

# 3 Main result

We prove that continuous generalized AdaDelta obeys the same tight rates for convergence of the loss and growth of the parameters as were established for plain gradient flow by Lyu & Li (2020, Corollary A.11), and also implicitly regularizes the parameters to converge in direction to a KKT point of a variant of the margin maximization problem in eq. (4), in which the objective function $\tfrac{1}{2}\|\boldsymbol{w}\|^2$ is replaced by the skewed $\tfrac{1}{2}\|\sqrt[4]{\frac{\boldsymbol{\delta}}{\boldsymbol{\varepsilon}}}\,\boldsymbol{w}\|^2$. If the quotient $\boldsymbol{\delta}/\boldsymbol{\varepsilon}$ of the numerical stability hyperparameters is isotropic (i.e. $\boldsymbol{\delta}_j/\boldsymbol{\varepsilon}_j$ are equal for all $j \in [p]$), then this modification does not alter the directions of the KKT points, so the implicit regularization is the same as was established for plain gradient flow by Lyu & Li (2020, Theorem A.8) and Ji & Telgarsky (2020, Theorem 3.1). Otherwise, the sets of points $\boldsymbol{w}$ that have the same objective value $\tfrac{1}{2}\|\sqrt[4]{\frac{\boldsymbol{\delta}}{\boldsymbol{\varepsilon}}}\,\boldsymbol{w}\|^2$ are ellipsoids which are not spheres, and the directions of the KKT points may be different from those for the margin maximization problem in eq. (4). Moreover, these conclusions are robust with respect to changing the exponential decay coefficients schedule $\boldsymbol{\rho}$ hyperparameter.

**Theorem 2.** *Under Assumptions 1 and 2, for the continuous generalized AdaDelta defined in Process 2, with either the exponential loss or the logistic loss, we have that $\mathcal{L}(\boldsymbol{w}(t)) = \Theta\left(\frac{1}{t(\log t)^{2-2/L}}\right)$, $\|\boldsymbol{w}(t)\| = \Theta((\log t)^{1/L})$, and $\boldsymbol{w}(t)$ converges in direction to a KKT point of the problem*

$$\text{minimize: } \tfrac{1}{2}\|\sqrt[4]{\tfrac{\boldsymbol{\delta}}{\boldsymbol{\varepsilon}}}\,\boldsymbol{w}\|^2 \qquad \text{subject to: } y^{(i)} \Phi(\boldsymbol{w}, \boldsymbol{x}^{(i)}) \geq 1 \text{ for all } i \in [n] \,.$$

*Remark* 3. The growth of the Euclidean length $\|\boldsymbol{w}(t)\|$ of the parameters to infinity during the training process is an artifact of the $L$-positively homogeneous predictor function and the strictly decreasing individual loss function (both exponential and logistic) — indeed, once perfect training accuracy is achieved, scaling the parameters $\boldsymbol{w}(t)$ by any $\alpha > 1$ decreases the total loss. Theorem 2 however establishes that this growth is slow, bounded above and below by the $L^{\text{th}}$ root of $\log t$. Note also that the convergence shown in Theorem 2 to a KKT point of the skewed normalized margin maximization problem is directional, and so independent of the growth of $\|\boldsymbol{w}(t)\|$.

Our proof of Theorem 2 builds on the following result, stated here in terms of the notations in this paper.

**Theorem 4** (Wang et al. (2021, Theorems 2, 3, and 10)). *Under Assumption 1, if $\boldsymbol{\chi} \in \mathbb{R}^p$ has no zero component and $\boldsymbol{v}, \boldsymbol{\beta} \colon [0, \infty) \to \mathbb{R}^p$ are arcs that obey the adaptive gradient flow*

$$\boldsymbol{v}'(t) = -\boldsymbol{\beta}(t)\,\widetilde{\partial}\widehat{\mathcal{L}}(\boldsymbol{v}(t)) \quad \text{for almost all } t \geq 0 \,,$$

*where $\widehat{\mathcal{L}}(\boldsymbol{v}) \coloneqq \mathcal{L}(\boldsymbol{\chi}\,\boldsymbol{v}) = \sum_{i=1}^n \ell\left(y^{(i)} \Phi(\boldsymbol{\chi}\,\boldsymbol{v}, \boldsymbol{x}^{(i)})\right)$ is with either exponential or logistic $\ell$, if $\widehat{\mathcal{L}}(\boldsymbol{v}(t_0)) < \ell(0)$ at some time $t_0$, if $\lim_{t \to \infty} \boldsymbol{\beta}(t) = \mathbf{1}$, and if $\mathrm{d}\log \boldsymbol{\beta}(t)/\mathrm{d}t$ is Lebesgue integrable, then we have that $\widehat{\mathcal{L}}(\boldsymbol{v}(t)) = \Theta\left(\frac{1}{t(\log t)^{2-2/L}}\right)$, $\|\boldsymbol{v}(t)\| = \Theta((\log t)^{1/L})$,[4] and $\boldsymbol{v}(t)$ converges in direction to a KKT point of the problem*

$$\text{minimize: } \tfrac{1}{2}\|\boldsymbol{v}\|^2 \qquad \text{subject to: } y^{(i)} \Phi(\boldsymbol{\chi}\,\boldsymbol{v}, \boldsymbol{x}^{(i)}) \geq 1 \text{ for all } i \in [n] \,.$$

To apply Theorem 4 to the continuous generalized AdaDelta in Process 2, we define for all $t \geq 0$:

$$\boldsymbol{\chi} \coloneqq \sqrt[4]{\tfrac{\boldsymbol{\varepsilon}}{\boldsymbol{\delta}}} \qquad \boldsymbol{v}(t) \coloneqq \boldsymbol{w}(t)/\boldsymbol{\chi} \qquad \boldsymbol{\beta}(t) \coloneqq \sqrt{\tfrac{1+\boldsymbol{h}(t)/\boldsymbol{\varepsilon}}{1+\boldsymbol{g}(t)/\boldsymbol{\delta}}} \qquad \widetilde{\partial}\widehat{\mathcal{L}}(\boldsymbol{v}(t)) \coloneqq \boldsymbol{\chi}\,\widetilde{\partial}\mathcal{L}(\boldsymbol{w}(t)) \,.$$

In terms of the *skewing vector* $\boldsymbol{\chi}$ and the *skewed adapter* $\boldsymbol{\beta}(t)$, eq. (3) that governs the parameters trajectory can then be restated as

$$\boldsymbol{w}'(t) = -\boldsymbol{\chi}^2\,\boldsymbol{\beta}(t)\,\widetilde{\partial}\mathcal{L}(\boldsymbol{w}(t)) \quad \text{for almost all } t \geq 0 \,. \tag{5}$$

Hence $\boldsymbol{v}'(t) = \boldsymbol{w}'(t)/\boldsymbol{\chi} = -\boldsymbol{\chi}\,\boldsymbol{\beta}(t)\,\widetilde{\partial}\mathcal{L}(\boldsymbol{w}(t)) = -\boldsymbol{\beta}(t)\,\widetilde{\partial}\widehat{\mathcal{L}}(\boldsymbol{v}(t))$ for almost all $t \geq 0$, as required. From Assumption 2.(iii), we have $\widehat{\mathcal{L}}(\boldsymbol{v}(t_0)) = \mathcal{L}(\boldsymbol{w}(t_0)) < \ell(0)$, also as required.

Therefore, to establish Theorem 2, it remains to prove that $\lim_{t \to \infty} \boldsymbol{\beta}(t) = \mathbf{1}$ and that $\mathrm{d}\log \boldsymbol{\beta}(t)/\mathrm{d}t$ is Lebesgue integrable. The remainder of this section consists of a sequence of lemmas, which culminates in Lemma 8 that shows the former fact and Lemma 9 that shows the latter fact. The lemmas in particular make use of Assumption 2.(ii) on the initialization of the arcs of the exponentially decaying averages $\boldsymbol{g}(t)$ and $\boldsymbol{h}(t)$, and of Assumption 2.(i) on the unboundedness of the integrals of the complements of the exponential decay coefficients, to show that each component of the skewed adapter $\boldsymbol{\beta}(t)$ monotonically (either from below or from above, depending on $\boldsymbol{g}(0)$ and $\boldsymbol{h}(0)$, and on the numerical stability terms $\boldsymbol{\delta}$ and $\boldsymbol{\varepsilon}$) tends to 1 as the time $t$ tends to infinity.

Our first lemma shows a useful expression for the derivative of $\mathbf{1}$ minus the squared skewed adapter.

**Lemma 5.** *For almost all $t \geq 0$ we have $\frac{\mathrm{d}}{\mathrm{d}t}(\mathbf{1} - \boldsymbol{\beta}(t)^2) = -\frac{\mathbf{1}-\boldsymbol{\rho}(t)}{\mathbf{1}+\boldsymbol{g}(t)/\boldsymbol{\delta}}(\mathbf{1} - \boldsymbol{\beta}(t)^2)$.*

*Proof.* Observe that

$$
\begin{aligned}
\frac{\mathrm{d}}{\mathrm{d}t}(\mathbf{1} - \boldsymbol{\beta}(t)^2) &= -\frac{\boldsymbol{h}'(t)/\boldsymbol{\varepsilon} + \boldsymbol{h}'(t)\,\boldsymbol{g}(t)/\boldsymbol{\varepsilon}\boldsymbol{\delta} - \boldsymbol{g}'(t)/\boldsymbol{\delta} - \boldsymbol{g}'(t)\,\boldsymbol{h}(t)/\boldsymbol{\delta}\boldsymbol{\varepsilon}}{(1 + \boldsymbol{g}(t)/\boldsymbol{\delta})^2} && \text{by the def. of } \boldsymbol{\beta}(t) \\[2mm]
&= -\frac{\mathbf{1}-\boldsymbol{\rho}(t)}{(1+\boldsymbol{g}(t)/\boldsymbol{\delta})^2}\left(\begin{array}{c} \boldsymbol{w}'(t)^2/\boldsymbol{\varepsilon} - \boldsymbol{h}(t)/\boldsymbol{\varepsilon} - \widetilde{\partial}\mathcal{L}(\boldsymbol{w}(t))^2/\boldsymbol{\delta} + \boldsymbol{g}(t)/\boldsymbol{\delta} \\ +\boldsymbol{w}'(t)^2\boldsymbol{g}(t)/\boldsymbol{\varepsilon}\boldsymbol{\delta} - \widetilde{\partial}\mathcal{L}(\boldsymbol{w}(t))^2\boldsymbol{h}(t)/\boldsymbol{\delta}\boldsymbol{\varepsilon} \end{array}\right) && \text{by eqs. (1) and (2)} \\[2mm]
&= -\frac{\mathbf{1}-\boldsymbol{\rho}(t)}{(1+\boldsymbol{g}(t)/\boldsymbol{\delta})^2}\left(\begin{array}{c} \boldsymbol{g}(t)/\boldsymbol{\delta} - \boldsymbol{h}(t)/\boldsymbol{\varepsilon} + (\boldsymbol{w}'(t)^2/\boldsymbol{\varepsilon})(1+\boldsymbol{g}(t)/\boldsymbol{\delta}) \\ -(\widetilde{\partial}\mathcal{L}(\boldsymbol{w}(t))^2/\boldsymbol{\delta})(1+\boldsymbol{h}(t)/\boldsymbol{\varepsilon}) \end{array}\right) && \text{rearranging} \\[2mm]
&= -\frac{\mathbf{1}-\boldsymbol{\rho}(t)}{(1+\boldsymbol{g}(t)/\boldsymbol{\delta})^2}(\boldsymbol{g}(t)/\boldsymbol{\delta} - \boldsymbol{h}(t)/\boldsymbol{\varepsilon}) && \text{by eq. (5)} \\[2mm]
&= -\frac{\mathbf{1}-\boldsymbol{\rho}(t)}{1+\boldsymbol{g}(t)/\boldsymbol{\delta}}(\mathbf{1} - \boldsymbol{\beta}(t)^2) && \text{calculation}
\end{aligned}
$$

for almost all $t \geq 0$. $\qquad\square$

The second lemma verifies that the exponentially decaying averages of squared gradients and squared adapted gradients are nonnegative at all times.

---

[4] Wang et al. (2021, Theorem 10) contains a typo: the bound $\Theta\left(\frac{1}{(\log t)^{1/L}}\right)$ should be $\Theta((\log t)^{1/L})$.

**Lemma 6.** *For all $t \geq 0$ we have $\boldsymbol{g}(t) \geq \boldsymbol{0}$ and $\boldsymbol{h}(t) \geq \boldsymbol{0}$.*

*Proof.* Suppose $\boldsymbol{g}(t)_j < 0$ for some $t \geq 0$ and $j \in [p]$. By Assumption 2.(ii) and since $\boldsymbol{g}_j \colon [0, \infty) \to \mathbb{R}$ is an arc, there exists $t_0 < t$ such that $\boldsymbol{g}(t_0)_j = 0$ and for all $\tau \in (t_0, t]$ we have $\boldsymbol{g}(\tau)_j < 0$. Recalling that $\boldsymbol{\rho}_j \colon [0, \infty) \to [0, 1]$, from eq. (1) we obtain that $\boldsymbol{g}'(\tau)_j \geq 0$ for almost all $\tau \in (t_0, t]$. Hence $\boldsymbol{g}(t)_j = \boldsymbol{g}(t_0)_j + \int_{t_0}^t \boldsymbol{g}'(\tau)_j \, \mathrm{d}\tau \geq 0$, which is a contradiction.

Supposing $\boldsymbol{h}(t)_j < 0$ for some $t \geq 0$ and $j \in [p]$, we obtain a contradiction analogously, using eq. (2). $\qquad\square$

Consider an arbitrary $j^{\text{th}}$ component of the skewed adapter. By Assumption 2.(ii), its initial value $\boldsymbol{\beta}(0)_j$ is positive. The proof of the third lemma shows that there are three cases: if $\boldsymbol{\beta}(0)_j < 1$ then $\boldsymbol{\beta}(t)_j$ is monotonically nondecreasing and bounded above by 1, if $\boldsymbol{\beta}(0)_j = 1$ then $\boldsymbol{\beta}(t)_j$ is constant, and if $\boldsymbol{\beta}(0)_j > 1$ then $\boldsymbol{\beta}(t)_j$ is monotonically nonincreasing and bounded below by 1.

**Lemma 7.** *The following hold for each $j \in [p]$.*

*(i) For almost all $t \geq 0$ we have: if $0 < \boldsymbol{\beta}(t)_j < 1$ then $\boldsymbol{\beta}'(t)_j \geq 0$, if $\boldsymbol{\beta}(t)_j = 1$ then $\boldsymbol{\beta}'(t)_j = 0$, and if $\boldsymbol{\beta}(t)_j > 1$ then $\boldsymbol{\beta}'(t)_j \leq 0$.*

*(ii) For all $t \geq 0$ we have $\boldsymbol{\beta}(t)_j \geq \min\{\boldsymbol{\beta}(0)_j, 1\}$.*

*Proof.* Consider an arbitrary $j \in [p]$.

For part (i), suppose $t \geq 0$ is such that Lemma 5 applies. If $0 < \boldsymbol{\beta}(t)_j < 1$ then $1 - \boldsymbol{\beta}(t)_j^2 > 0$, so from Lemmas 5 and 6, and by recalling that $0 \leq \boldsymbol{\rho}(t)_j \leq 1$, we have $\mathrm{d}(1 - \boldsymbol{\beta}(t)_j^2)/\mathrm{d}t \leq 0$, i.e. $\mathrm{d}\boldsymbol{\beta}(t)_j^2/\mathrm{d}t \geq 0$. But $\mathrm{d}\boldsymbol{\beta}(t)_j^2/\mathrm{d}t = 2\boldsymbol{\beta}(t)_j\boldsymbol{\beta}'(t)_j$, and so $\boldsymbol{\beta}'(t)_j \geq 0$. In the remaining two cases, namely if $\boldsymbol{\beta}(t)_j = 1$ or $\boldsymbol{\beta}(t)_j > 1$, the reasoning is analogous.

Now part (ii) follows by part (i) and the fact that, since $\boldsymbol{g}_j, \boldsymbol{h}_j \colon [0, \infty) \to \mathbb{R}$ are arcs, so is $\boldsymbol{\beta}_j$. $\qquad\square$

Lemma 7 provides a lower bound for the skewed adapter, but it leaves open its asymptotic behavior. The next lemma fills that gap, establishing that all its components converge to 1. This equivalently means that the continuous generalized AdaDelta adapter $\sqrt{\frac{\boldsymbol{\varepsilon}+\boldsymbol{h}(t)}{\boldsymbol{\delta}+\boldsymbol{g}(t)}}$ converges to the squared skewing vector $\boldsymbol{\chi}^2 = \sqrt{\frac{\boldsymbol{\varepsilon}}{\boldsymbol{\delta}}}$.

**Lemma 8.** *We have that $\lim_{t \to \infty} \boldsymbol{\beta}(t) = \boldsymbol{1}$.*

*Proof.* We first use Lemma 7 to prove that the squared gradients have bounded integrals. To that end, for all $t \geq 0$ we have

$$\mathcal{L}(\boldsymbol{w}(0)) > \mathcal{L}(\boldsymbol{w}(0)) - \mathcal{L}(\boldsymbol{w}(t)) \qquad\qquad \text{since } \mathcal{L}(\boldsymbol{w}(t)) > 0$$

$$= -\int_0^t (\mathrm{d}\mathcal{L}(\boldsymbol{w}(\tau))/\mathrm{d}\tau)\mathrm{d}\tau \qquad\qquad \text{calculus}$$

$$= -\int_0^t \langle \widetilde{\partial}\mathcal{L}(\boldsymbol{w}(\tau)), \boldsymbol{w}'(\tau)\rangle \mathrm{d}\tau \qquad\qquad \text{by Proposition 1}$$

$$= \int_0^t \langle \widetilde{\partial}\mathcal{L}(\boldsymbol{w}(\tau)), \sqrt{\tfrac{\boldsymbol{\varepsilon}}{\boldsymbol{\delta}}}\,\boldsymbol{\beta}(\tau)\,\widetilde{\partial}\mathcal{L}(\boldsymbol{w}(\tau))\rangle \mathrm{d}\tau \qquad\qquad \text{by eq. (5)}$$

$$\geq \int_0^t \langle \sqrt{\tfrac{\boldsymbol{\varepsilon}}{\boldsymbol{\delta}}}\min\{\boldsymbol{\beta}(0), \boldsymbol{1}\}, \widetilde{\partial}\mathcal{L}(\boldsymbol{w}(\tau))^2\rangle \mathrm{d}\tau \qquad\qquad \text{by Lemma 7.(ii)}$$

$$= \sum_{j \in [p]} \sqrt{\tfrac{\boldsymbol{\varepsilon}_j}{\boldsymbol{\delta}_j}}\min\{\boldsymbol{\beta}(0)_j, 1\} \int_0^t \widetilde{\partial}\mathcal{L}(\boldsymbol{w}(\tau))_j^2 \, \mathrm{d}\tau \qquad\qquad \text{rearranging ,}$$

so for each $j \in [p]$ we have

$$\int_0^t \widetilde{\partial}\mathcal{L}(\boldsymbol{w}(\tau))_j^2 \, \mathrm{d}\tau < \sqrt{\tfrac{\boldsymbol{\delta}_j}{\boldsymbol{\varepsilon}_j}}\frac{\mathcal{L}(\boldsymbol{w}(0))}{\min\{\boldsymbol{\beta}(0)_j, 1\}} =: C_j \;.$$

Now suppose $j \in [p]$. Equation (1) and Lemma 6 imply that

$$\boldsymbol{g}(t)_j = \boldsymbol{g}(0)_j + \int_0^t \boldsymbol{g}'(\tau)_j \, \mathrm{d}\tau \leq \boldsymbol{g}(0)_j + \int_0^t \widetilde{\partial}\mathcal{L}(\boldsymbol{w}(\tau))_j^2 \, \mathrm{d}\tau < \boldsymbol{g}(0)_j + C_j =: C_j^{\dagger} \quad \text{for all } t \geq 0 \,,$$

so recalling Lemma 5 and Lemma 7.(i), and setting $C_j^{\ddagger} := \frac{1}{1+C_j^{\dagger}/\boldsymbol{\delta}_j}$, we have

$$\mathrm{d}\log|1 - \boldsymbol{\beta}(t)_j^2|/\mathrm{d}t \leq -C_j^{\ddagger}(1 - \boldsymbol{\rho}(t)_j) \quad \text{for almost all } t \geq 0 \text{ such that } \boldsymbol{\beta}(t)_j \neq 1 \,.$$

Hence

$$|1 - \boldsymbol{\beta}(t)_j^2| \leq |1 - \boldsymbol{\beta}(0)_j^2| \exp\left(-C_j^{\ddagger} \int_0^t (1 - \boldsymbol{\rho}(\tau)_j)\mathrm{d}\tau\right) \quad \text{for all } t \geq 0 \,,$$

which by positivity of $C_j^{\ddagger}$ and unboundedness of the integrals of the complements of the exponential decay coefficients (cf. Assumption 2.(i)) establishes the lemma. $\qquad\square$

Our final lemma shows that the components of the skewed adapter converge without large fluctuations of their logarithms.

**Lemma 9.** *The function* $\mathrm{d}\log\boldsymbol{\beta}(t)/\mathrm{d}t$ *is Lebesgue integrable.*

*Proof.* For each $j \in [p]$, from the proof of Lemma 7 we have that $\mathrm{d}\log\boldsymbol{\beta}(t)_j/\mathrm{d}t = \boldsymbol{\beta}'(t)_j/\boldsymbol{\beta}(t)_j$ is either nonnegative for almost all $t \geq 0$, or nonpositive for almost all $t \geq 0$. Thus by Lemma 8 we have

$$\int_0^\infty |\mathrm{d}\log\boldsymbol{\beta}(t)/\mathrm{d}t|\mathrm{d}t = \left|\int_0^\infty (\mathrm{d}\log\boldsymbol{\beta}(t)/\mathrm{d}t)\mathrm{d}t\right| = |\log \boldsymbol{1} - \log\boldsymbol{\beta}(0)| = |\log\boldsymbol{\beta}(0)| < \boldsymbol{\infty} \,. \qquad\square$$

## 4    Experiments

To test our theoretical result for generalized AdaDelta, we evaluated comparatively five algorithms:

**SGD:** Stochastic gradient descent as implemented in PyTorch[5].

**AdaDelta:** Its standard PyTorch implementation[3], with the exponential decay coefficient $\varrho = 0.9$, and the numerical stability term $\epsilon = 10^{-5}$.

**AdaDeltaS:** This is AdaDelta amended to have the exponential decay coefficients follow the schedule $\varrho_k = 1 - 0.1/(1 + \lfloor 100k/K \rfloor)$, where $K$ is the total number of steps. Thus $1 - \varrho_k$ follows a harmonic sequence, increasing the coefficient from $\varrho_0 = 0.9$ at the first step to $\varrho_{K-1} = 0.999$ at the last step, which lessens aggressiveness of the decay in computing the averages of the squared gradients and the squared adapted gradients along the training.

**AdaDeltaN:** This is AdaDelta amended to have the numerical stability terms different in the numerator and the denominator of the adaptor, and different across the network parameters. At the start of the training, each component of $\boldsymbol{\delta}$ and of $\boldsymbol{\varepsilon}$ is sampled independently from $10^{-5+X}$, where $X$ is a centered Gaussian with standard deviation 1 for the smaller two networks we consider, and with standard deviation 0.25 for the largest network.

**AdaDeltaNS:** This combines the extensions in AdaDeltaS and AdaDeltaN, i.e. has exponential decay coefficients that follow the specified schedule as well as numerical stability terms whose components are initialized randomly as above.

We performed experiments in the following three gradually more complex settings. The total compute for the perceptron setting was around 10min on a mid-range CPU; whereas one run of all five algorithms for the smaller convolutional setting took roughly 2h, and for the larger convolutional setting took roughly 12h, in both cases on a mid-range GPU.

---

[5]https://pytorch.org/docs/stable/generated/torch.optim.SGD.html

**Two-layer Leaky ReLU perceptron on a two-dimensional dataset.** Following Wang et al. (2021), we first considered a perceptron with parameters $v \in \mathbb{R}$ and $\boldsymbol{w} \in \mathbb{R}^2$, whose prediction for an input $\boldsymbol{x} \in \mathbb{R}^2$ equals $v\,\sigma(\langle \boldsymbol{w}, \boldsymbol{x} \rangle)$, where $\sigma$ is the Leaky ReLU nonlinearity with inactive gradient 0.5.

The dataset $\left\{(\boldsymbol{x}^{(1)}, y^{(1)}), \ldots, (\boldsymbol{x}^{(100)}, y^{(100)})\right\}$ consists of 50 points sampled from $(\cos 0.5, \sin 0.5) + \boldsymbol{u}$ independently and labelled 1, and 50 points sampled from $-(\cos 0.5, \sin 0.5) + \boldsymbol{u}$ independently and labelled $-1$, where $\boldsymbol{u}$ is distributed uniformly on the square $[-0.6, 0.6]^2$.

This toy setting is convenient for three-dimensional visualizations of the adapter's reciprocal square roots. We trained the network for $K = 5000$ full-batch epochs using the exponential loss, with learning rate 0.1 for SGD and learning rate 1 for the four variants of AdaDelta. The learning rate for SGD was chosen so that it achieves perfect training accuracy after a similar number of epochs as the four versions of AdaDelta. We repeated the training 100 times, where in each round the five algorithms used the same randomly initialized parameters and numerical stability terms (if applicable). The results are shown in fig. 1.

We observe that:

- all rounds achieve 100% training accuracy by around 256 epochs, in line with the separation Assumption 2.(iii);

- the final normalized margin, which here equals $\min_{i \in [100]} y^{(i)}\, v_K\, \sigma(\langle \boldsymbol{w}_K, \boldsymbol{x}^{(i)} \rangle)/(v_K^2 + \|\boldsymbol{w}_K\|^2)$, is within a higher and much narrower range for SGD and with isotropic numerical stability hyperparameters (i.e. for AdaDelta and AdaDeltaS), confirming the prediction of Theorem 2 that, without the isotropy, the implicit regularization may not be towards margin maximization;

- the final adapter's reciprocal square root, which in the notations of Procedure 1 equals $\sqrt[4]{\frac{\boldsymbol{\delta} + \boldsymbol{g}_K}{\boldsymbol{\varepsilon} + \boldsymbol{h}_{K-1}}}$, has a large variance of its direction when the numerical stability hyperparameters have random components (i.e. for AdaDeltaN and AdaDeltaNS), and it is the limit of this direction that according to the proof of Theorem 2 determines the nature of the implicit regularization.

**Four-layer convolutional network on MNIST.** Also following Wang et al. (2021), we then considered the network from Mądry, Makelov, Schmidt, Tsipras, and Vladu (2018) with biases removed, and in order consisting of: 32-channel $5 \times 5$-filter convolutional, ReLU, 2-kernel 2-stride max-pooling, 64-channel $3 \times 3$-filter convolutional, ReLU, 2-kernel 2-stride max-pooling, 1024-width fully connected, ReLU, and 10-width fully connected.

We trained the network on MNIST (LeCun, Bottou, Bengio, and Haffner, 1998) from the default PyTorch random initialization for 500 epochs using the cross-entropy loss: in a finer regime with batch size 100, learning rate 0.01 for SGD, and learning rate 0.1 for the four variants of AdaDelta; and in a coarser regime with batch size 1000, learning rate 0.1 for SGD, and learning rate 1 for the four variants of AdaDelta. The learning rates for SGD were chosen so that it achieves perfect training accuracy after a similar number of epochs as the four versions of AdaDelta. The results are shown in fig. 2.

For both regimes, we observe that:

- all algorithms achieve perfect training accuracy by around 250 epochs, in line with the separation Assumption 2.(iii);

- the normalized margin, whose values here are relatively small partly due to the division by the fourth power of the norm of the network parameters, grows significantly higher for SGD and with isotropic numerical stability hyperparameters (i.e. for AdaDelta and AdaDeltaS), confirming the prediction of Theorem 2 that, without the isotropy, the implicit regularization may not be towards margin maximization;

- after all algorithms achieve perfect training accuracy, their test accuracies remain roughly constant, in the finer regime without significant differences among the four versions of AdaDelta, and in the coarser regime with AdaDelta and AdaDeltaS generalizing slightly but consistently better;

- the training loss shrinks faster when the numerical stability hyperparameters have random components (i.e. for AdaDeltaN and AdaDeltaNS), however remains within a constant multiplicative band of the training loss when those hyperparameters are isotropic;

- the results are not substantially affected by whether the exponential decay coefficients follow the increasing schedule, i.e. they are similar for AdaDelta and AdaDeltaS, and also for AdaDeltaN and AdaDeltaNS.

**VGG on CIFAR-10.**   Following Lyu & Li (2020), we finally considered the 14-layer VGG-16 (Simonyan and Zisserman, 2015) with biases only at the first level, and in order consisting of: 64-channel convolutional then ReLU, repeated 2 times; max-pooling; 128-channel convolutional then ReLU, repeated 2 times; max-pooling; 256-channel convolutional then ReLU, repeated 3 times; max-pooling; 512-channel convolutional then ReLU, repeated 3 times; max-pooling; 256-channel convolutional then ReLU, repeated 3 times; max-pooling; 10-width fully connected. Each convolutional layer is $3 \times 3$-filter, and each max-pooling is 2-kernel 2-stride.

We trained the network on CIFAR-10 (Krizhevsky, 2009) from the default PyTorch random initialization for 1000 epochs using the cross-entropy loss: in a finer regime with batch size 100, and learning rate 0.1 for all five algorithms; and in a coarser regime with batch size 250, and learning rate 0.25 for all five algorithms. The results are shown in fig. 3.

For both regimes, we observe that:

- all algorithms achieve perfect training accuracy by around 250 epochs, in line with the separation Assumption 2.(iii);

- the normalized margin, whose values here are relatively small partly due to the division by the $14^{\text{th}}$ power of the network norm, grows significantly higher for SGD and with isotropic numerical stability hyperparameters, with the exception in the coarser regime of AdaDelta whose training loss shrinks considerably slower compared with AdaDeltaS — this is in line with Theorem 2, whose prediction of implicit regularization towards margin maximization depends both on the isotropy and on convergence to zero training loss;

- after all algorithms achieve perfect training accuracy, their test accuracies remain roughly constant and without substantial differences among them (approximately within $81 \pm 1\%$ in the finer regime, or $79 \pm 1\%$ in the coarser regime), with the exception in the coarser regime of AdaDelta and AdaDeltaN whose training losses shrink considerably slower compared with AdaDeltaS and AdaDeltaNS;

- whether the exponential decay coefficients follow the increasing schedule does not substantially affect the results in the finer regime, however in the courser regime AdaDeltaS and AdaDeltaNS shrink the training loss considerably faster than AdaDelta and AdaDeltaN, leading to higher normalized margins as well as better generalization.

## 5  Conclusion

Our main result, Theorem 2 which holds under Assumptions 1 and 2, indicates that AdaDelta may be used in practical methods that rely on the optimization algorithm implicitly regularizing the network parameters to converge in direction to a margin maximization KKT point, which places AdaDelta in the same category as e.g. RMSProp and Adam without momentum (Wang et al., 2021, Theorem 7). This relies on the isotropy of the quotient of the numerical stability hyperparameters, without which according to Theorem 2 the implicit regularization of AdaDelta may not be towards margin maximization, as is the case for e.g. AdaGrad (Wang et al., 2021, Theorem 6).

Relaxing Assumption 1 on the predictor function is challenging even without considering adaptive algorithms, e.g. the definability excludes pathological examples such as based on the "Mexican hat" function (cf. Lyu & Li (2020, Appendix J)).

We focused on binary classification for simplicity of presentation; it is straightforward to extend Theorem 2 for logistic loss to an arbitrary number of classes (cf. Wang et al. (2021, Appendix E)).

An important future goal is to obtain a counterpart of Theorem 2 directly for adaptive gradient descent as in Procedure 1. This is also a challenge already without adaptivity: the directional convergence result (Ji

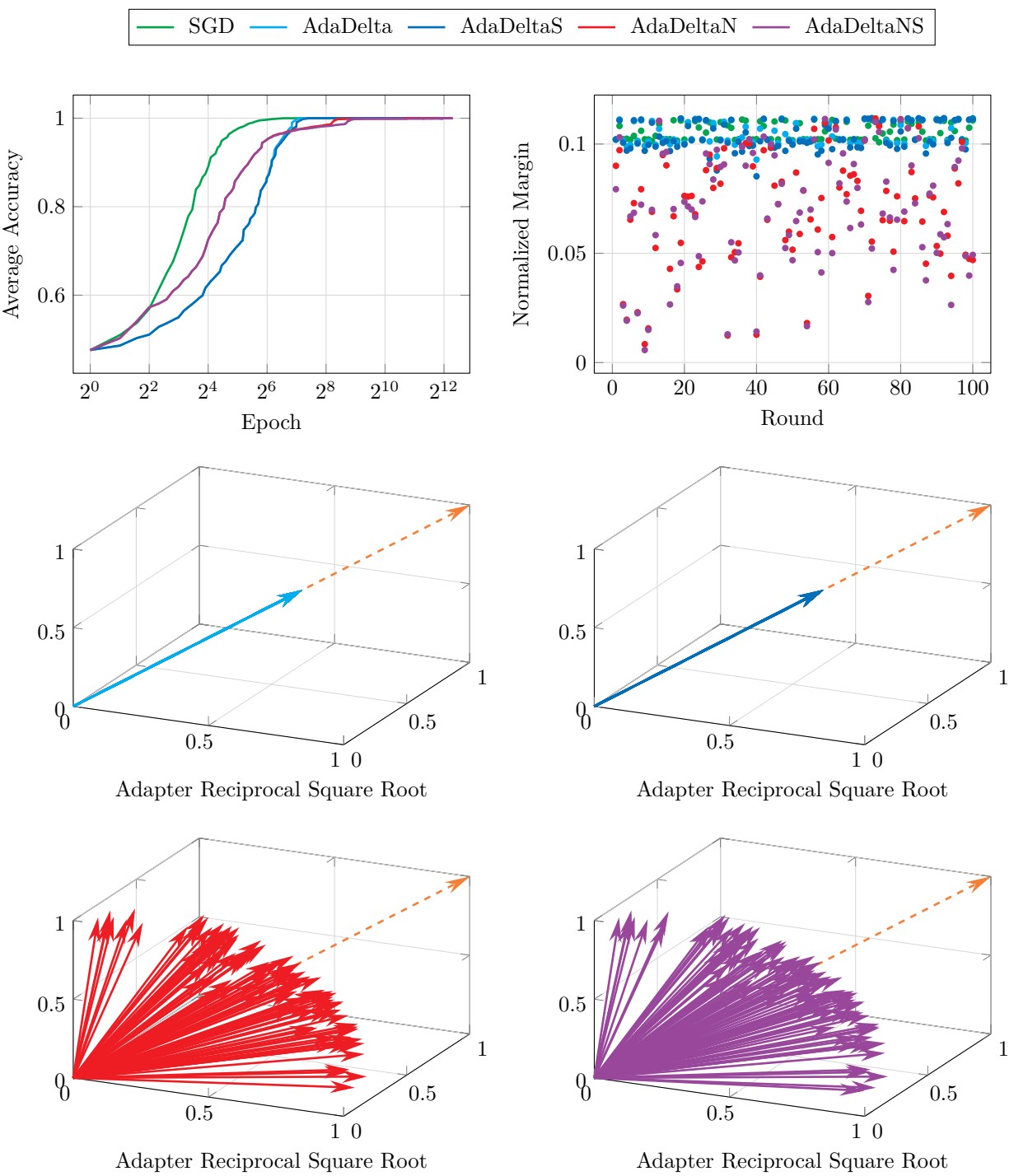

Figure 1: We trained a two-layer Leaky ReLU perceptron on a synthetic binary classification dataset which consists of 100 points in two-dimensional space. Each of 100 rounds of the experiment consisted of randomly initializing the network parameters and the numerical stability terms (if applicable), and then running separately each of the five algorithms for 5000 epochs. The plots in the top row show: on the left, the training accuracy averaged over the rounds, depending on the epoch; and on the right, the normalized margin at the end of the training, across each of the rounds. The plots in the next two rows visualize the direction of the adapter's reciprocal square root at the end of the training, where the isotropic direction is indicated by the longer dashed orange arrow.

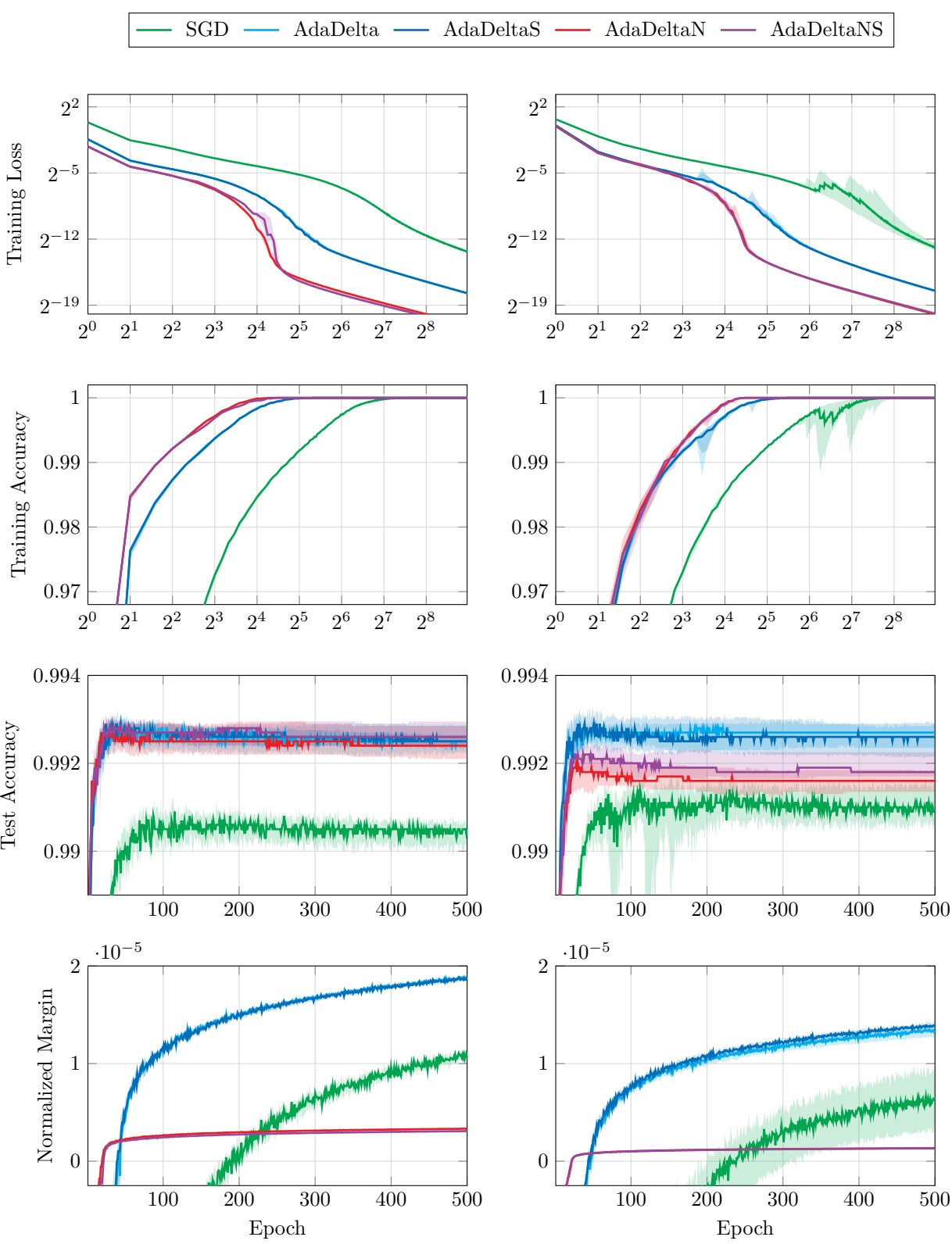

Figure 2: The results of training a 4-layer convolutional network on MNIST, with batch sizes and learning rates for the right-hand column that are 10 times larger than for the left-hand column. The solid lines show the median values, and the shaded areas are between the 25th and 75th percentiles, over 19 runs.

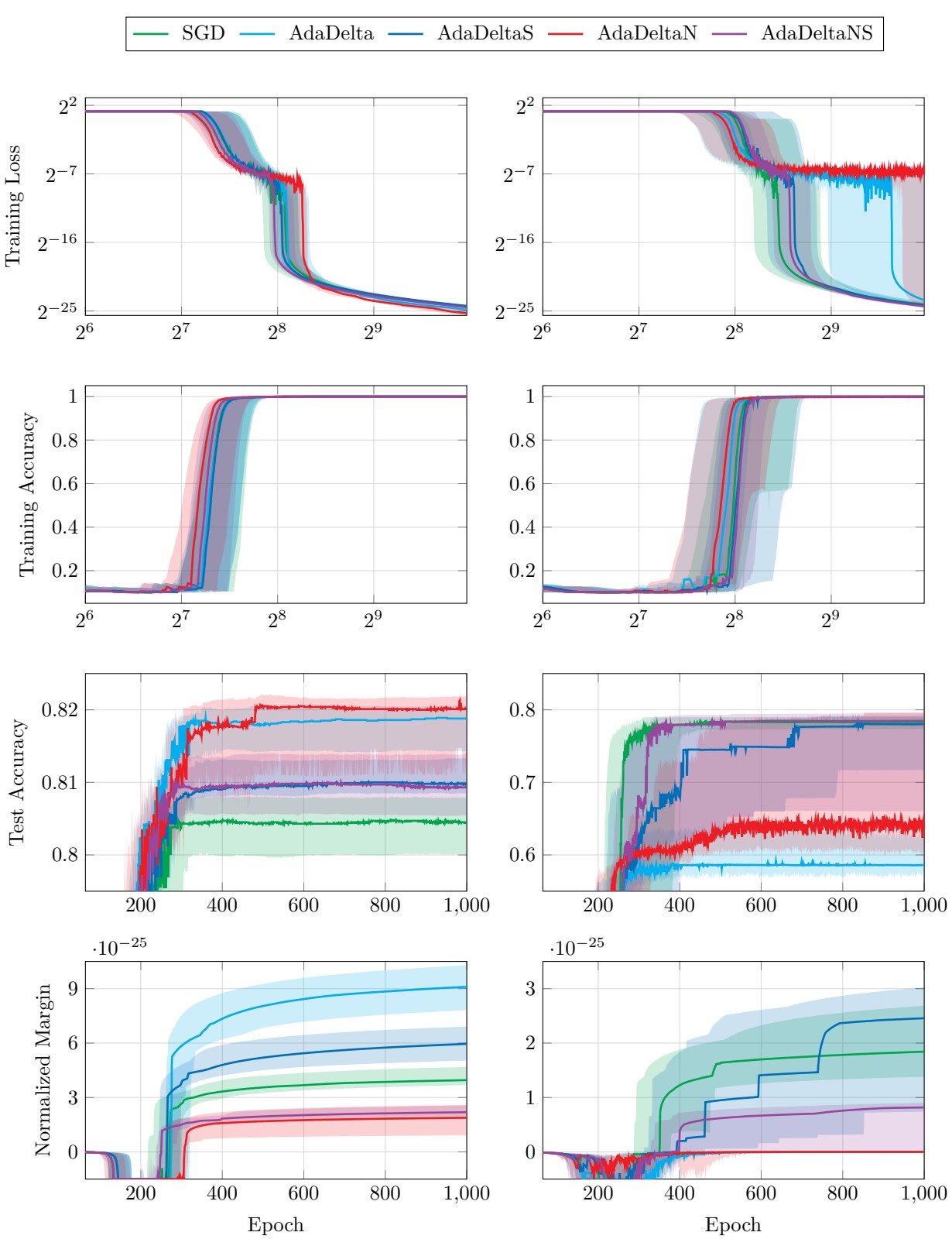

Figure 3: The results of training a 14-layer convolutional network on CIFAR-10, with batch sizes and learning rates for the right-hand column that are 2.5 times larger than for the left-hand column. The solid lines show the median values, and the shaded areas are between the 25th and 75th percentiles, over 19 runs.

& Telgarsky, 2020, Theorem 3.1) is only for gradient flow, and the characterization of directional limits for gradient descent (Lyu & Li, 2020, Theorem E.3) assumes $\mathcal{C}^2$-smoothness of the predictor function which excludes nonlinearities such as ReLU and Leaky ReLU.

**Broader Impact Statement**

This is foundational research on a general-purpose algorithm for optimizing neural networks. Greater understanding of its implicit regularization properties may lead to machine learning models that have cheaper training, greater efficiency, and increased performance.

**Acknowledgments**

We acknowledge the Centre for Discrete Mathematics and Its Applications at the University of Warwick for partial support.

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
