# OpenReview forum: "Implicit Regularization of AdaDelta"
_TMLR — Accepted by TMLR_

### Review · Reviewer_wbBq · 2024-08-29

**Summary Of Contributions:**

This paper analyzes the implicit regularization phenomenon, i.e. the tendency of some optimization algorithm to converge towards solutions with good generalization properties, for the AdaDelta algorithm and some of its variants. The broader setting is fitting a variety of neural networks for a classification problem, with either exponential or logistic loss.
It builds on several previous results for different algorithms (and namely The Implicit Bias for Adaptive Optimization Algorithms on Homogeneous Neural Networks, Wang et al., 2021) to provide a rigorous theoretical analysis and proof of the conditions where the AdaDelta method converges towards a KKT point of a specific margin maximization problem. It then illustrates the implicit regularization properties of the different variants of AdaDelta on various network types and datasets, ranging from generated data to CIFAR-10.

**Audience:**

Yes

**Claims And Evidence:**

Yes

**Requested Changes:**

See weaknesses section above. In short:
- Add more structure to the proof, e.g. with a proof sketch section where the link between each step of the proof is clearly explained
- Add more details in the conclusion, restate the Wang et al. results and show how they connect to what was just shown
- Discuss experiments in more details and give more insights
- Add an an appendix with code and illustration for reproducibility

**Strengths And Weaknesses:**

Strengths:
- The proof is rigorous and provides a solid theoretical grounding for implicit regularization with AdaDelta, which was never explored before
- Results are solid and well corroborated by experiments
- Theoretical and empirical analyses are extensive and provide interesting insights not only on AdaDelta, but also on what can happen when some conditions are not present, e.g. when quotient of the numerical stability parameters is not isotropic

Weaknesses:
- Clarity: structure of the proof is confusing, the paper does not always outline very well the link between different lemmas - the reasoning behind the different parts of the proof is not always easy to follow
- Conclusion of the proof is unclear: the paper does not explain well how what was just shown connects with Theorems 2, 3 and 10 of the Wang et al. paper
- Empirical results could be discussed in more details: the discussions per dataset are fairly short, and having 3 pages of graphs with practically no comment is hard to follow

---

> ### Author Response · Authors · 2024-11-04
> **Thank you very much for the insightful review.**
>
> > Clarity: structure of the proof is confusing, the paper does not always outline very well the link between different lemmas - the reasoning behind the different parts of the proof is not always easy to follow
>
> > Conclusion of the proof is unclear: the paper does not explain well how what was just shown connects with Theorems 2, 3 and 10 of the Wang et al. paper
>
> > Add more structure to the proof, e.g. with a proof sketch section where the link between each step of the proof is clearly explained
>
> > Add more details in the conclusion, restate the Wang et al. results and show how they connect to what was just shown
>
> We have reorganized and expanded the proof.  It now starts with a restatement of the Wang et al. results, after which we clarify what remains to be proved, and sketch how the sequence of lemmas that follows will accomplish that.  We have split Lemma 4 into two (now Lemmas 6 and 7), provided more explanation for what Lemmas 4 and 5 show (before the statements of now Lemmas 7 and 8), and provided more details in the proofs of Lemmas 4 and 5 (now Lemmas 6, 7, and 8).
>
> > Empirical results could be discussed in more details: the discussions per dataset are fairly short, and having 3 pages of graphs with practically no comment is hard to follow
>
> > Discuss experiments in more details and give more insights
>
> We have significantly expanded the Experiments section, particularly the discussion of the VGG on CIFAR-10 setting.  We have also revised the plots, which for the two more involved settings now show the cumulative results of 19 runs per each experiment.
>
> > Add an an appendix with code and illustration for reproducibility
>
> We have now provided the code in the supplementary materials, which includes instructions on how to run it.

---

### Review · Reviewer_ECdm · 2024-09-25

**Summary Of Contributions:**

This work focuses on a specific adaptive gradient method called AdaDelta. The paper shows that the continuous-time version of AdaDelta (in the limit of small step sizes) is an adaptive gradient flow in the sense of the work of Wang et al. 2021, i.e. a preconditioned gradient flow with a preconditioner whose coordinates converge to 1 with a Lebesgue integrable derivative of its logarithm. Using this result, they apply a theorem established in Wang et al. 2021 to deduce their main result: In continuous time, AdaDelta enjoys the same implicit regularization as the vanilla gradient flow which was shown in prior work. More precisely, for positively homogeneous  locally Lipschitz and definable neural networks with suitable regularity and technical assumptions, they show that the weights of the neural network updated with AdaDelta converge in direction to a KKT point of a maximum margin problem where the objective (squared weight norm) is slightly distorted by the numerical stability hyper parameters of AdaDelta. In particular, this work considers exponential decay functions varying with time at a controlled speed by an integrability assumption. Numerical experiments on the popular MNIST and CIFAR-10 datasets with convolutional networks support and illustrate part of the main theoretical result.

**Audience:**

Yes

**Broader Impact Concerns:**

Does not apply.

**Claims And Evidence:**

Yes

**Requested Changes:**

Please see the above section for questions and comments which I believe deserve to be addressed. The main points that are essential are the technical aspects as well as some of the questions regarding experiments and the alignment of the results with the main theoretical findings. The rest of the comments are also important for strengthening the paper and efforts from the authors to address them to the best of their ability will be appreciated, especially concerning the benefit of AdaDelta over vanilla gradient flow and possible results in discrete time.

Some additional comments and suggestions for improvement below:

- To better highlight the motivation, I suggest to further recall and insist on the existing correlation results between max-margin maximization and generalization error.

- As the work mainly relies on prior work for deriving the main result (Theorem 2), giving slightly more expanded discussion about their result (including some intuitions and key arguments to establish it) would be appreciated to make the paper more self-contained.

- Right after Theorem 2, you start deriving some Lemmas for analyzing $\beta(t)$. It would be better to guide the reader from the beginning about your proof strategy. I had to check the reference in more details to understand why you are proving the lemmas. I would rather start by stating the general proof strategy (rather than finishing the proof with it), saying that you rewrite the flow as an adaptive gradient flow in the sense of Wang et al. 21 and that you would like to prove that $\beta(t)$ goes to 1 and that the derivative of the adapter’s log function is Lebesgue integrable, you can even state their result for completeness.

**Strengths And Weaknesses:**

**Strengths:**
- The general topic of generalization is important in deep learning. This work builds on (and contributes to) a well-established line of research investigating the relationships between optimization and generalization, specifically for the popular class of adaptive methods which is very successful in practice.
- The exposition and writing are clear overall. The introduction for the topic of implicit regularization, the goal and the contributions of the paper are quite clear.
- I checked the proofs and they are correct to the best of my knowledge, up to some specific concerns/questions described in details below.

**Weaknesses:** The following comments include a discussion about weaknesses as well as comments and questions for clarifications.

- The main result regarding implicit regularization is only shown for the continuous-time AdaDelta and no result is established for the (discrete-time) algorithm of interest:

— I understand that prior work seems to be mostly restricted to continuous time as briefly mentioned in the conclusion. Nevertheless, AdaDelta corresponds to $\eta = 1$ which makes it a particular case of Procedure 1. However, if you take $\eta \to 0$ to obtain the continuous-time system, why would the obtained flow be a relevant description of AdaDelta?

— Is there any reason why the authors do not derive a result similar to Theorem 8 in Wang et al. 2021 to have at least a result for the actual algorithm (in discrete time) even if it is at the price of stronger smoothness assumptions? Since the main result of the present work builds on results of Wang et al. 2021, I am wondering why not also including such a result in discrete time. Is there any major technical issue? If you assume $C^2$ smoothness of the predictor function, can you still state a result characterizing the directional limits for your adaptive AdaDelta flow? (similarly to e.g. Theorem E.3 of Lyu and Li ’20 that you mention in the conclusion).


- Theorem 2:

— Could you comment more on the relevance of this implicit regularization result compared to vanilla gradient flow? What does the result suggest in practice: Should we always use $\delta = \epsilon$ to asymptotically match gradient flow? Are there any specific differences with vanilla gradient flow that provide some kind of advantage to AdaDelta (or adaptive gradient flow more generally)? Any theoretical or actionable practical implementation insight regarding hyperparameters? It seems that only numerical stability hyperparameters might play a role asymptotically and affect implicit regularization.

— As a follow-up to the previous comment, one would expect some difference with respect to gradient flow depending on the specific choice of the exponential decay coefficient rate for instance that should effect the provided rates. Indeed, the rate of convergence of $\beta(t)$ to 1 seems to be driven by two ingredients: a) how fast $\rho(t)$ goes to 1 as captured by the integrability condition Assumption 2 (i) and b) how fast $\tilde{\partial} \mathcal{L}(\omega(t))$ goes to zero as captured by the constant $C_j$ appearing in the square integrability condition that you show in Lemma 5, i.e. convergence to `stationarity’.
The current analysis does not exhibit such a difference as it shows overall that both adaptive and vanilla gradient flows have the same implicit regularization properties including for rates. It would be interesting to further comment (if possible) on how these convergence rates might be beneficial for the convergence of the weights compared to standard gradient flow, i.e.  if a different analysis taking this into account could be possible.

— The norm of the the weights diverges to infinity at a log rate for $t \to \infty$, what’s the meaning of this blow-up result? Can you provide any intuition for clarification (perhaps based on prior work also showing the same result)?

-- Any additional comment about the comparison to other adaptive algorithms previously studied in the literature (e.g. RMSProp) in theory and in practice?

- About assumption 1:

— I guess the exponential function is excluded due to positive homogeneity, activation functions such as the sigmoid are also excluded as a consequence. Does it mean that only (almost) `linear’ networks (composition of linear layers with potentially Relu, … zeroing out inputs in some ranges) are allowed? As the O-minimal class of functions considered is  chosen to contain the exponential function (as if it was indeed an advantage), this is rather confusing.

— Comment about $L$ for positive homogeneity: I guess it is related here to the number of layers. A remark would be useful to clarify.

- The so-called separability assumption (Assumption 2 (iii)) is formulated a bit differently in Wang et al. 2021 (see their assumption 1. III). Do both formulations coincide exactly? Can you comment on your formulation here?

- last sentence of the conclusion: `the benefits of anisotropic numerical stability terms and of exponential decay coefficients that we observed in some experiments’. Can you be more specific here? Numerical stability hyperparameters are usually introduced to avoid blowing up or vanishing numerical issues. It seems odd to use them for a different purpose and I am not expecting this to provide any uniform advantage over SGD for all problems. The analysis provided shows that using $\delta \neq \epsilon$ only leads asymptotically to a rescaling of the (vanilla) gradient flow by the constant square root of $\delta$ over $\epsilon$ (given that $\beta$ goes to 1). This can be simply seen as a rescaling of the objective (loss) function by a constant.

- While the discussion is relatively clear regarding the very specific topic of implicit regularization, the related work discussion is quite minimal regarding continuous-time analysis of adaptive gradient methods. While the main focus of this work is rather on implicit regularization, the analysis of the continuous-time system is the crucial technical contribution of this work enabling to establish implicit regularization. There are a number of works on the topic in the literature that might even include AdaDelta as a particular case (Barakat and Bianchi (SIOPT 2021) for Adam, Da Silva and Gazeau (JMLR 2021) for a class of adaptive methods in continuous time, Barakat et al. (EJS 2022) for a class of adaptive methods in both continuous and discrete time, Gadat and Gavra (JMLR 2022) for RMSProp and AdaGrad in both continuous and discrete time, more recently Xiao et al. (JMLR 2024) for adaptive methods for nonsmooth optimization). Notice for instance that several similar technical steps (e.g. Lemma 4, 5 and their proofs, exponential decaying coefficient integrability conditions such as Assumption 2 (i) …) were also shown and used in this related literature for a larger class of adaptive methods including also momentum algorithms (although for different final purposes). Note also that some of these works also deal with time varying exponential decay coefficients and use similar assumptions. It might even be that a general unified analysis including momentum (which seems to be missing in the literature) might be conducted based on these works and ideas from the present work and prior work (Wang et al. 2021) to show implicit regularization for a large class of adaptive gradient methods in a unified manner.

A. B. Da Silva, M. Gazeau. A general system of differential equations to model first-order adaptive algorithms, JMLR 2020.

A. Barakat, P. Bianchi. Convergence and dynamical behavior of the Adam algorithm for non-convex stochastic optimization, SIOPT 2021.

A. Barakat, P. Bianchi, W. Hachem, Sh. Schechtman. Stochastic optimization with momentum: convergence, fluctuations, and traps avoidance, EJS 2021.

S. Gadat, I. Gavra. Asymptotic study of stochastic adaptive algorithms in non-convex landscape, JMLR 2022.

N. Xiao, X. Hu, X. Liu, K-C. Toh. Adam-family methods for nonsmooth optimization with convergence guarantees, JMLR 2024.
- Technical analysis:

— Please justify the first inequality in the proof of Lemma 5 precisely. It seems that you should require or show that at least $\mathcal{L}(\omega(t))$ should admit a limit when $t \to \infty$ for the integral from 0 to infinity of the derivative to exist. This would mean that the training loss needs to stabilize asymptotically, is this an assumption or do you rather prove it somewhere? It seems that this limit is proven to be equal to zero for RMSProp in Wang et al. 2021 in the second step of their proof strategy (see discussion sec. 5 p. 6 and Lemma 4 p. 8). Maybe using a similar argument to Lemma 12 in Wang et al. ’21 should fix this. Please clarify.

— In the proof of Lemma 4, the non negativity of $g$ can be shown more clearly. While the main arguments are there, the proof deserves a clearer presentation: e.g. suppose there exists $t_0 > 0$ s.t. $g(t_0) < 0$ then there should exist $t_1 <  t_0$ s.t. $g(t_1) = 0$ (supposing $g$ is positive at $0$) by continuity of $g$ but then $g’(t_1)$ is nonnegative by (1) and this would contradict the sign of $g(t_0) < 0$. Please clarify your reasoning. In particular, is $g$ necessarily continuous in your nonsmooth setting, given that there might be some jumps in the (Clarke) sub gradients of the loss function?

— Proof of Lemma 5, p. 7: Please add more details about how you derive the inequality. Writing $g(t) = g(0) + \int_0^t g’(s) ds$ and injecting $g’(s)$ in eq. (1) does not immediately lead to the result. It seems that you are also using the nonnegativity of $g$ here (as seems to be proved in the proof of Lemma 4, see previous item discussion) and the non negativity of $\rho$.

- Experiments:

— Why considering two different standard deviations depending on the size of the networks in the end of p. 7?

— Is the learning rate tuned for SGD? How?

— p. 9: `the test accuracy is consistently high for AdaDelta and AdaDeltaS, corroborating the link between the normalized margin and generalization’. Does this mean that AdaDelta does generalize better than SGD given Fig. 2 (c) (at least in this example)? This seems to be contradicting some of the empirical results reported in the introduction regarding generalization of adaptive methods (Wilson et al. 2017).

— Fig. 1 p. 10: It seems that the S variants of the algorithm do not make any difference. I guess this is related to the way the exponential decay coefficients are set. Are the results sensitive to this choice. In particular, what do we observe for faster or slower increasing exponential decay coefficients towards 1?

— What do all the different vectors represent in Fig. (f), (g)? Are these final adapter’s reciprocal square roots across different rounds with different random numerical stability hyperparameters?

— Fig. 2 (c)-(d): Large margin does not seem to always translate to better generalization. Any comment?

— The normalized margins in Fig. 2. (d) (same in Fig. 3 (d)) do not seem to converge to the same value for SGD, AdaDelta and AdaDeltaS unlike what is predicted by the theory. Is this expected, consistent? Any explanation?


More minor:
- p. 1: `if the adapter of an algorithm without momentum …’: on a first reading (without going through the whole paper), it is not very clear what is meant by an adapter here. Although you provide a reference, a clearer formulation (exempting the reader from checking the reference) would be appreciated.
- p. 2: `RMSProp and Adam without momentum’: to the best of my knowledge Adam without momentum and RMSProp as defined in their respective original papers/lectures are the same, are you maybe referring to some other variants of them? Please clarify.
- Typos: a few occurrences of AdaDeta instead of AdaDelta.
- p. 6: please avoid overloading the numerator in the second equation, I understand that this is more compact but this can be confusing at first reading, maybe put in factor the denominator instead to gain some space.
- It seems that you are also showing in the proof of Lemma 4 that $\beta$ is actually monotone, that is a result that might be highlighted in the lemma.
- p. 7: `the skewed adapter converge without large fluctuations’: fluctuations of $\log \beta(t)$ here I guess, it might be that $\beta(t)$ actually fluctuates quite a lot when translating back from the log scale.

---

> ### Author Response · Authors · 2024-11-04
> **Thank you very much for the impressively thorough review.**
>
> > The main result regarding implicit regularization is only shown for the continuous-time AdaDelta and no result is established for the (discrete-time) algorithm of interest:
>
> > — I understand that prior work seems to be mostly restricted to continuous time as briefly mentioned in the conclusion. Nevertheless, AdaDelta corresponds to $\eta = 1$ which makes it a particular case of Procedure 1. However, if you take $\eta \to 0$ to obtain the continuous-time system, why would the obtained flow be a relevant description of AdaDelta?
>
> > — Is there any reason why the authors do not derive a result similar to Theorem 8 in Wang et al. 2021 to have at least a result for the actual algorithm (in discrete time) even if it is at the price of stronger smoothness assumptions? Since the main result of the present work builds on results of Wang et al. 2021, I am wondering why not also including such a result in discrete time. Is there any major technical issue? If you assume $C^2$ smoothness of the predictor function, can you still state a result characterizing the directional limits for your adaptive AdaDelta flow? (similarly to e.g. Theorem E.3 of Lyu and Li ’20 that you mention in the conclusion).
>
> We agree that characterizing the implicit bias in discrete time is an important question, and we highlight it as such in the Conclusion.  Even with assuming $C^2$ smoothness of the predictor function, and with seeking only to characterize the directional limits (i.e. without showing convergence), this seems challenging to us.  We remark that Theorem 8 in Wang et al. 2021 applies only to AdaGrad, RMSProp, and Adam without momentum, and their proof of it as far as we can see is specific to those adaptive algorithms, which differ from AdaDelta in that they do not involve an exponentially decaying average in the numerator of the adapter.  (In contrast, their Theorems 2, 3, and 10, which are for continuous time, and on which our work is based, apply to a class of adaptive algorithms with convergent adapters, and we prove that AdaDelta belongs to that class.)
>
> We also remark that the standard PyTorch implementation of AdaDelta has a tunable learning rate, i.e. $1$ is only its default value.
>
> > Theorem 2:
>
> > — Could you comment more on the relevance of this implicit regularization result compared to vanilla gradient flow? What does the result suggest in practice: Should we always use $\delta = \varepsilon$ to asymptotically match gradient flow? Are there any specific differences with vanilla gradient flow that provide some kind of advantage to AdaDelta (or adaptive gradient flow more generally)? Any theoretical or actionable practical implementation insight regarding hyperparameters? It seems that only numerical stability hyperparameters might play a role asymptotically and affect implicit regularization.
>
> Yes, our main result can be seen as confirming that the implicit regularization of AdaDelta is the same as that of vanilla gradient flow, and also making clear what key role the numerical stability hyperparameters play in that.  We have expanded the start of the Conclusion to highlight that.
>
> In a nutshell, our message to practitioners is: if you are using AdaDelta to speed up your optimization, then our theorem provides some confidence that you are not doing so to the detriment of the margin (which is in contrast to e.g. AdaGrad); and also some of our experiments suggest that if your training loss is struggling to decrease beyond some value, then you may want to try scheduling the exponential decay coefficient to increase gradually towards $1$.
>
> > — As a follow-up to the previous comment, one would expect some difference with respect to gradient flow depending on the specific choice of the exponential decay coefficient rate for instance that should effect the provided rates. Indeed, the rate of convergence of $\beta(t)$ to $1$ seems to be driven by two ingredients: a) how fast $\rho(t)$ goes to $1$ as captured by the integrability condition Assumption 2 (i) and b) how fast $\tilde{\partial} \mathcal{L}(w(t))$ goes to zero as captured by the constant $C_j$ appearing in the square integrability condition that you show in Lemma 5, i.e. convergence to `stationarity’. The current analysis does not exhibit such a difference as it shows overall that both adaptive and vanilla gradient flows have the same implicit regularization properties including for rates. It would be interesting to further comment (if possible) on how these convergence rates might be beneficial for the convergence of the weights compared to standard gradient flow, i.e. if a different analysis taking this into account could be possible.
>
> This could be explored in future work.  The only comment we currently have is that the rate of convergence of $\beta(t)$ to $1$ seems unlikely to impact significantly the rates of convergence of the weights as it only affects the learning rate coefficients by factors close to $1$.

---

> ### Author Response · Authors · 2024-11-04
> **Part 2 of our response.**
>
> > Theorem 2:
>
> > — The norm of the weights diverges to infinity at a log rate for $t \to \infty$, what’s the meaning of this blow-up result? Can you provide any intuition for clarification (perhaps based on prior work also showing the same result)?
>
> We have added comments about this immediately after the statement of the theorem.
>
> > — Any additional comment about the comparison to other adaptive algorithms previously studied in the literature (e.g. RMSProp) in theory and in practice?
>
> We have expanded the start of the Conclusion to compare to what was known about the implicit regularization of RMSProp, Adam without momentum, and AdaGrad.
>
> > About assumption 1:
>
> > — I guess the exponential function is excluded due to positive homogeneity, activation functions such as the sigmoid are also excluded as a consequence. Does it mean that only (almost) `linear’ networks (composition of linear layers with potentially Relu, … zeroing out inputs in some ranges) are allowed? As the O-minimal class of functions considered is chosen to contain the exponential function (as if it was indeed an advantage), this is rather confusing.
>
> > — Comment about $L$ for positive homogeneity: I guess it is related here to the number of layers. A remark would be useful to clarify.
>
> Thank you for these points.  We have expanded the discussion of Assumption 1 to include examples of admissible activation functions with polynomials, to clarify the role of the definability of the exponential function (it is needed for the loss functions), and to comment further on how the homogeneity exponent $L$ is related to the number of layers (especially in the presence of activation functions with polynomials).
>
> > The so-called separability assumption (Assumption 2 (iii)) is formulated a bit differently in Wang et al. 2021 (see their assumption 1. III). Do both formulations coincide exactly? Can you comment on your formulation here?
>
> They coincide exactly, and we have now explained that in the discussion of Assumption 2.
>
> > last sentence of the conclusion: `the benefits of anisotropic numerical stability terms and of exponential decay coefficients that we observed in some experiments’. Can you be more specific here? Numerical stability hyperparameters are usually introduced to avoid blowing up or vanishing numerical issues. It seems odd to use them for a different purpose and I am not expecting this to provide any uniform advantage over SGD for all problems. The analysis provided shows that using $\delta \neq \varepsilon$ only leads asymptotically to a rescaling of the (vanilla) gradient flow by the constant square root of $\delta$ over $\varepsilon$ (given that $\beta$ goes to $1$). This can be simply seen as a rescaling of the objective (loss) function by a constant.
>
> We think you are right.  We have removed this paragraph from the Conclusion as it was possibly too speculative, however we have revised our comments about this in the Experiments section, in discussion of the four-layer convolutional network on MNIST.
>
> > While the discussion is relatively clear regarding the very specific topic of implicit regularization, the related work discussion is quite minimal regarding continuous-time analysis of adaptive gradient methods. While the main focus of this work is rather on implicit regularization, the analysis of the continuous-time system is the crucial technical contribution of this work enabling to establish implicit regularization. There are a number of works on the topic in the literature that might even include AdaDelta as a particular case (Barakat and Bianchi (SIOPT 2021) for Adam, Da Silva and Gazeau (JMLR 2021) for a class of adaptive methods in continuous time, Barakat et al. (EJS 2022) for a class of adaptive methods in both continuous and discrete time, Gadat and Gavra (JMLR 2022) for RMSProp and AdaGrad in both continuous and discrete time, more recently Xiao et al. (JMLR 2024) for adaptive methods for nonsmooth optimization). Notice for instance that several similar technical steps (e.g. Lemma 4, 5 and their proofs, exponential decaying coefficient integrability conditions such as Assumption 2 (i) …) were also shown and used in this related literature for a larger class of adaptive methods including also momentum algorithms (although for different final purposes). Note also that some of these works also deal with time varying exponential decay coefficients and use similar assumptions. It might even be that a general unified analysis including momentum (which seems to be missing in the literature) might be conducted based on these works and ideas from the present work and prior work (Wang et al. 2021) to show implicit regularization for a large class of adaptive gradient methods in a unified manner.
>
> Thank you for highlighting these works, we have now included them in the discussion of further related work in the Introduction.

---

> ### Author Response · Authors · 2024-11-04
> **Part 3 of our response.**
>
> > Technical analysis:
>
> > — Please justify the first inequality in the proof of Lemma 5 precisely. It seems that you should require or show that at least $\mathcal{L}(w(t))$ should admit a limit when $t \to \infty$ for the integral from $0$ to infinity of the derivative to exist. This would mean that the training loss needs to stabilize asymptotically, is this an assumption or do you rather prove it somewhere? It seems that this limit is proven to be equal to zero for RMSProp in Wang et al. 2021 in the second step of their proof strategy (see discussion sec. 5 p. 6 and Lemma 4 p. 8). Maybe using a similar argument to Lemma 12 in Wang et al. ’21 should fix this. Please clarify.
>
> > — In the proof of Lemma 4, the non negativity of $g$ can be shown more clearly. While the main arguments are there, the proof deserves a clearer presentation: e.g. suppose there exists $t_0 > 0$ s.t. $g(t_0) < 0$ then there should exist $t_1 < t_0$ s.t. $g(t_1) = 0$ (supposing $g$ is positive at $0$) by continuity of $g$ but then $g'(t_1)$ is nonnegative by (1) and this would contradict the sign of $g(t_0) < 0$. Please clarify your reasoning. In particular, is $g$ necessarily continuous in your nonsmooth setting, given that there might be some jumps in the (Clarke) sub gradients of the loss function?
>
> > — Proof of Lemma 5, p. 7: Please add more details about how you derive the inequality. Writing $g(t) = g(0) + \int_0^t g'(s) ds$ and injecting $g'(s)$ in eq. (1) does not immediately lead to the result. It seems that you are also using the nonnegativity of $g$ here (as seems to be proved in the proof of Lemma 4, see previous item discussion) and the non negativity of $\rho$.
>
> Thank you for these detailed suggestions, we have expanded the proofs of Lemmas 4 and 5 (now Lemmas 6, 7, and 8) accordingly.
>
> > Experiments:
>
> > — Why considering two different standard deviations depending on the size of the networks in the end of p. 7?
>
> The larger network ($14$ layers) struggles with optimization if the standard deviation is much larger.  The differences of $\pm 1$ in these exponents can cause the limits of the components of the skewed adapter to differ by a factor of $100$.
>
> > — Is the learning rate tuned for SGD? How?
>
> We have now remarked about this, it was chosen so that SGD achieves perfect training accuracy after a similar number of
> epochs as the four versions of AdaDelta.
>
> > — p. 9: `the test accuracy is consistently high for AdaDelta and AdaDeltaS, corroborating the link between the normalized margin and generalization’. Does this mean that AdaDelta does generalize better than SGD given Fig. 2 (c) (at least in this example)? This seems to be contradicting some of the empirical results reported in the introduction regarding generalization of adaptive methods (Wilson et al. 2017).
>
> Our main point here is that AdaDelta and AdaDeltaS generalize better than AdaDeltaN and AdaDeltaNS.  Our main theorem suggests that the implicit regularization of the former pair is the same as for SGD, and that the implicit regularization of AdaDeltaN and AdaDeltaNS is different.
>
> > — Fig. 1 p. 10: It seems that the S variants of the algorithm do not make any difference. I guess this is related to the way the exponential decay coefficients are set. Are the results sensitive to this choice. In particular, what do we observe for faster or slower increasing exponential decay coefficients towards 1?
>
> We apologise, there was a bug related to the exponential decay coefficients in the experiments for fig. 1.  We have corrected it, and the updated plots show a greater difference of the S variants.
>
> > — What do all the different vectors represent in Fig. (f), (g)? Are these final adapter’s reciprocal square roots across different rounds with different random numerical stability hyperparameters?
>
> Yes, exactly.  We explain this at the end of the figure caption.
>
> > — Fig. 2 (c)-(d): Large margin does not seem to always translate to better generalization. Any comment?
>
> Indeed, e.g. the relatively large empirical study by Jiang et al. in ICLR 2020 in the best cases shows that they correlate more or less strongly.
>
> > — The normalized margins in Fig. 2. (d) (same in Fig. 3 (d)) do not seem to converge to the same value for SGD, AdaDelta and AdaDeltaS unlike what is predicted by the theory. Is this expected, consistent? Any explanation?
>
> We suggest the theory does not necessarily predict global maximization of the normalized margin, only directional convergence to some KKT point of the problem.

---

> ### Author Response · Authors · 2024-11-04
> **Part 4 of our response.**
>
> > More minor:
>
> > p. 1: `if the adapter of an algorithm without momentum …’: on a first reading (without going through the whole paper), it is not very clear what is meant by an adapter here. Although you provide a reference, a clearer formulation (exempting the reader from checking the reference) would be appreciated.
>
> > p. 2: `RMSProp and Adam without momentum’: to the best of my knowledge Adam without momentum and RMSProp as defined in their respective original papers/lectures are the same, are you maybe referring to some other variants of them? Please clarify.
>
> > Typos: a few occurrences of AdaDeta instead of AdaDelta.
>
> > p. 6: please avoid overloading the numerator in the second equation, I understand that this is more compact but this can be confusing at first reading, maybe put in factor the denominator instead to gain some space.
>
> > It seems that you are also showing in the proof of Lemma 4 that $\beta$ is actually monotone, that is a result that might be highlighted in the lemma.
>
> > p. 7: `the skewed adapter converge without large fluctuations’: fluctuations of $\log \beta(t)$ here I guess, it might be that $\beta(t)$ actually fluctuates quite a lot when translating back from the log scale.
>
> Thank you, we have revised the paper to take care of all these points.
>
> > To better highlight the motivation, I suggest to further recall and insist on the existing correlation results between max-margin maximization and generalization error.
>
> We have now pointed to several works on understanding generalization using margin-based quantities at the end of the Introduction.
>
> > As the work mainly relies on prior work for deriving the main result (Theorem 2), giving slightly more expanded discussion about their result (including some intuitions and key arguments to establish it) would be appreciated to make the paper more self-contained.
>
> > Right after Theorem 2, you start deriving some Lemmas for analyzing $\beta(t)$. It would be better to guide the reader from the beginning about your proof strategy. I had to check the reference in more details to understand why you are proving the lemmas. I would rather start by stating the general proof strategy (rather than finishing the proof with it), saying that you rewrite the flow as an adaptive gradient flow in the sense of Wang et al. 21 and that you would like to prove that $\beta(t)$ goes to $1$ and that the derivative of the adapter’s log function is Lebesgue integrable, you can even state their result for completeness.
>
> Thank you for these suggestions, we have now reorganized the proof of the main result so that it starts with a restatement of the results of Wang et al. that we build on, and an outline of our consequent proof strategy.

---

### Review · Reviewer_uzSm · 2024-10-14

**Summary Of Contributions:**

This paper investigates the implicit regularization of the AdaDelta optimizer for training a class of neural networks for solving classification problems. The concept of implicit regularization is one of the tools that helps in explaining the generalization behavior of gradient-based optimizers. The contribution lies in a generalization (in terms of non-uniform numerical stability parameters and learning rate scheduling) of the existing AdaDelta algorithm: whereas implicit regularization has been theoretically studied before for few other adaptive gradient optimizers (AdaGrad, RMSProp, Adam without momentum), this is probably the first time such analysis is carried out when both numerator and denominator of $\Delta x$ have exponentially moving averages. The theoretical guarantees are numerically verified with a set of experiments.

**Audience:**

Yes

**Broader Impact Concerns:**

The broader impact that is evident - first time implicit regularization analysis when both numerator and denominator in an adaptive gradient algorithm have exponentially moving averages.

But I do not agree with the statement that this is a foundational research on a `general algorithm' for optimizing neural networks. While the research is foundational, it is limited in its current form only to a generalization (in terms of numerical stability and learning-rate schedule) of AdaDelta algorithm. It is not clear how the results extend to other algorithms for training neural networks, such as Adam and its variants. It is also not clear how the results will help in cheaper training and better efficiency, since the experimental results lack comparison with other optimizers on benchmark models.

**Claims And Evidence:**

Yes

**Requested Changes:**

Not Critical for Acceptance Recommdenation:
1. Explain the reason for introducing $\epsilon$ for better numerical stability. Possibly with instances where with $\epsilon=0$, AdaDelta can go unstable. This is not critical for acceptance, as the analysis still has value without $\epsilon$.
2. Is the result in Theorem 2 valid for both exponential loss and logistic loss. Nowhere in the theorem statement and in the proof, the loss function class is mentioned. Please mention it explicitly in or right before the theorem statement.

Critical:
1. Please check if the proofs of Lemma 4 and 5 can be simplified from my earlier comment. This is critical, as the paper is mostly theoretical.
2. Why does AdaDeltaNS have better test accuracy in Figure 2, despite not having the best margin maximzation?
3. The effect of numerical stablitity on AdaDelta and AdaDeltaN are also not consistent: AdaDeltaN is worse than AdaDelta, while AdaDeltaNS is better than AdaDeltaS.

**Strengths And Weaknesses:**

Strength:
1. This is probably the first time implicit regularization analysis is carried out when both numerator and denominator of $\Delta x$ in an adaptive gradient algorithm have exponentially moving averages.
2. The amount of implicit regularization has been characterized directly in terms of the algorithm parameters. The key theoretical tool that the authors used is Lemma 3, which does the heavy lifting in presence of exponentially moving average of both numerator and denominator of $\Delta x$.
3. The proofs in Section 3 are technically correct and well-presented.
4. The numerical results supports the main theoretical result.

Weakness/Limitation:
1. In $\Delta x$ of AdaDelta, the current gradient $\partial L (w_t)$ and the adaptive term $\beta(t)$ are separable. In Adam, the numerator of $\Delta x$ has exponentially moving sum of gradients, and hence, no separable gradient term $\partial L (w_t)$.  Thus, extending the presented analysis technique to such adaptive algorithms might be challenging.
2. Why is the numerical stability ($\epsilon$) required in the numerator? While the generalization in terms of $\eta$ and $\delta$ are meaningful, the purpose of  introducing $\epsilon$ is not clear.
3. Does not the statements of Lemma 4 and 5 follow directly from Lemma 3? Let us denote $y(t) = 1 - \beta(t)^2$. Then, Lemma 3 states that $\frac{dy(t)}{y(t)} = - dt * z(t)$, where $z(t)$ is a positive valued function of t. From this and by integration, we can easily get Lemma 4 and 5?
4. Figure 2: The qualitative behavior between AdaDeltaN and AdaDeltaNS are not consistent across training and testing. Also, test accuracy of AdaDeltaNS is the best, but normalized margin AdaDelta/AdaDeltaS is best. Is not normalized margin a characterization of generalization ability?

---

> ### Author Response · Authors · 2024-11-04
> **Thank you very much for the insightful review.**
>
> > In $\Delta x$ of AdaDelta, the current gradient $\partial L(w_t)$ and the adaptive term $\beta(t)$ are separable. In Adam, the numerator of $\Delta_x$ has exponentially moving sum of gradients, and hence, no separable gradient term $\partial L(w_t)$. Thus, extending the presented analysis technique to such adaptive algorithms might be challenging.
>
> You are right, and we agree that this might be challenging.
>
> > Why is the numerical stability ($\varepsilon$) required in the numerator? While the generalization in terms of $\eta$ and $\delta$ are meaningful, the purpose of introducing $\varepsilon$ is not clear.
>
> > Explain the reason for introducing $\varepsilon$ for better numerical stability. Possibly with instances where with $\varepsilon = 0$, AdaDelta can go unstable. This is not critical for acceptance, as the analysis still has value without $\varepsilon$.
>
> The numerical stability term in the numerator is a feature of standard AdaDelta (and by default equals $10^{-6}$, please see https://pytorch.org/docs/stable/generated/torch.optim.Adadelta.html).  Our understanding is that it helps keep the learning rate coefficients in the adapter away from zero.
>
> > Does not the statements of Lemma 4 and 5 follow directly from Lemma 3? Let us denote $y(t) = 1 - \beta(t)^2$. Then, Lemma 3 states that $\frac{d y(t)}{y(t)} = -dt * z(t)$, where $z(t)$ is a positive valued function of $t$. From this and by integration, we can easily get Lemma 4 and 5?
>
> > Please check if the proofs of Lemma 4 and 5 can be simplified from my earlier comment. This is critical, as the paper is mostly theoretical.
>
> We tried and did not succeed to shortcut the proofs along these lines, one issue is to show that $z(t)$ does not decrease to zero too quickly.  On the other hand, what you outline is a key component, which we use towards the end of the proof of Lemma 5 (now Lemma 8).
>
> > Figure 2: The qualitative behavior between AdaDeltaN and AdaDeltaNS are not consistent across training and testing. Also, test accuracy of AdaDeltaNS is the best, but normalized margin AdaDelta/AdaDeltaS is best. Is not normalized margin a characterization of generalization ability?
>
> > Why does AdaDeltaNS have better test accuracy in Figure 2, despite not having the best margin maximzation?
>
> > The effect of numerical stablitity on AdaDelta and AdaDeltaN are also not consistent: AdaDeltaN is worse than AdaDelta, while AdaDeltaNS is better than AdaDeltaS.
>
> We have redone the plots to reflect a greater number of runs of the experiments (now 19 for each plot in Figures 2 and 3), and updated the discussion in the Experiments section.  The issue with AdaDeltaNS in Figure 2 has basically disappeared, apparently it was an artifact of only 5 runs.
>
> We also remark that even e.g. the relatively large empirical study by Jiang et al. in ICLR 2020 in the best cases only demonstrates that normalized margin and generalization ability correlate more or less strongly.
>
> > Is the result in Theorem 2 valid for both exponential loss and logistic loss. Nowhere in the theorem statement and in the proof, the loss function class is mentioned. Please mention it explicitly in or right before the theorem statement.
>
> Thank you, we have now stated this explicitly in Theorem 2.
>
> > But I do not agree with the statement that this is a foundational research on a `general algorithm' for optimizing neural networks. While the research is foundational, it is limited in its current form only to a generalization (in terms of numerical stability and learning-rate schedule) of AdaDelta algorithm. It is not clear how the results extend to other algorithms for training neural networks, such as Adam and its variants.
>
> We agree, and we have reworded the Broader Impact Statement to clarify that AdaDelta is a general purpose algorithm, in the sense that it is in principle applicable to any neural network training task.
>
> > It is also not clear how the results will help in cheaper training and better efficiency, since the experimental results lack comparison with other optimizers on benchmark models.
>
> As we remark in the Introduction, AdaDelta is known to perform better sometimes than several other optimizers (please see e.g. Ruder 2016).  Our main result provides some confidence that, if margin maximization is a part of the goal, then one might use AdaDelta for faster training without jeopardizing the normalized margin, or one might use AdaDelta on a smaller model to achieve a comparable normalized margin than perhaps with another optimizer on a larger model.  However, indeed this paper is mostly theoretical, and our experiments primarily test Theorem 2 rather than constitute an extensive empirical study of AdaDelta.

---

### Decision · Action_Editor_S3My · 2024-11-28

**Recommendation:** Accept as is

**Comment:**

The papers claims now clearly match their content, there is an audience who would be interested, and revised version of the paper is already satisfactory.

**Audience:**

This will be mostly of interests to the community who wish to understand generalization of optimization methods, in particular, those who wish to analyse adaptive gradient methods in continuous time.

**Claims And Evidence:**

The paper analyses the continuous time dynamics of AdaDelta. The authors show that this continuous time dynamics enjoys a certain implicit regularization, by which they mean it converges to a KKT point that solves the margin maximization problem.

During the review process the papers proofs and structure has also greatly improved, making it easier to check their claims.